



# Changes in the annual sea ice freeze-thaw cycle in the Arctic Ocean from 2001 to 2018

Long Lin[1,2], Ruibo Lei[1*], Mario Hoppmann[3], Donald K. Perovich[4], Hailun He[2,5]

[1]Key Laboratory of Polar Science, MNR, Polar Research Institute of China, Shanghai, China;

[2]State Key Laboratory of Satellite Ocean Environment Dynamics, Second Institute of Oceanography, MNR, Hangzhou, China;

[3] Alfred-Wegener-Institut Helmholtz-Zentrum für Polar- und Meeresforschung, Bremerhaven, Germany

[4]Thayer School of Engineering, Dartmouth College, Dartmouth, NH, USA

[5]Southern Marine Science and Engineering Guangdong Laboratory (Zhuhai), Zhuhai, China;

*Correspondence to*: Ruibo Lei (leiruibo@pric.org.cn)

**Abstract.** The annual sea ice freeze-thaw cycle plays a crucial role in the Arctic atmosphere-ice-ocean system, regulating the seasonal energy balance of sea ice and the underlying surface ocean. Previous studies of the sea ice freeze-thaw cycle were often based on limited accessible in situ or easily available remotely sensed observations from surface. To better understand the responses of the sea ice to climate change and its coupling to the upper ocean, we combine measurements of the ice surface and bottom using multisource data to investigate the temporal and spatial variations in the freeze-thaw cycle of Arctic sea ice. Observations by 69 sea ice mass balance buoys (IMBs) collected from 2001 to 2018 revealed that the average ice basal melt onset in the Beaufort Gyre occurred on 23 May (±6 d), approximately 17 days earlier than the surface melt onset. And the average ice basal melt onset in the Central Arctic Ocean occurred on 17 June (±9 d), which was comparable with the surface melt onset. This inconsistency was mainly attributed to the difference in the seasonal variations of oceanic heat available to sea ice melt between the two regions. The overall average onset of basal ice growth of the pan Arctic Ocean occurred on 14 November, lagging approximately 3 months behind the surface freeze onset. This temporal delay was caused by a combination of the heat released from sea ice cooling, a heat flux from the ocean mixed layer and subsurface layer, as well as the thermal buffering of snow atop the ice. In the Beaufort Gyre region, both (Lagrangian) IMB observations (2001–2018) and (Eulerian) moored upward looking sonar (ULS) observations (2003–2018) revealed an earlier trend for basal melt onset, mainly linked to earlier warming of surface ocean. An earlier trend of basal ice growth was also identified from the IMB observations (multiyear ice), which we attributed to the overall reduction of ice thickness. In contrast, a delayed trend of basal ice growth was identified from the ULS observations, which was explained by the fact that the ice cover melted almost entirely by the end of summer in the recent years.



## 1 Introduction

Seasonal thermodynamic freezing and thawing processes are crucial to control the mass budget of the cryosphere (Planck et al., 2020; Derksen et al., 2012). In the Arctic Ocean, the presence of sea ice greatly modifies the exchanges of heat, momentum, and mass between the atmosphere and the ocean. The timings of the sea ice melt and freeze onsets (MO and FO), as well as the length of the melt and freeze seasons play a key role in the heat budget of the atmosphere-ice-ocean system, for example by altering the surface albedo and meltwater budget in summer (Perovich and Polashenski, 2012; Stroeve et al., 2014) and by brine discharge in both winter (Ivanov et al., 2016) and summer (Tian et al., 2018) through different mechanisms. Changes in the lengths of the melt and freeze seasons also regulate the degree of consolidation and mechanical strength of the sea ice field, and consequently enhance or weaken the mobility and deformation of the sea ice, even if the wind forcing does not change (Rampal et al., 2019; Lei et al., 2021). Passive microwave (PMW) satellite observations indicated that the length of the sea ice surface melt season is extending with a rate of 5 d decade$^{-1}$ due to both earlier MO and later FO, especially in the peripheral seas where seasonal sea ice is becoming dominant (Stroeve et al., 2014). Lengthening of the melt season leads to more solar energy absorption and storage in the ice-ocean system (Perovich et al., 2011), contributing to the thinning and loss of Arctic sea ice in summer (Perovich et al., 2015), promoting the bloom of ice algae and phytoplankton under the ice (Ardyna and Arrigo, 2020), and suppressing the ice recovery in winter (Timmermans, 2015; Ricker et al., 2021). Thus, the MO and FO can be considered as significant phenological indices of the Arctic climate system.

The majority of related studies have derived the phenological indices of the Arctic MO and FO based on observations of the ice surface (hereafter refer as SMO and SFO), such as in situ, reanalyzed or remotely-sensed near-surface air temperature (Rigor et al., 2000; Bliss and Anderson, 2018), passive microwave brightness temperature (Markus et al., 2009; Stroeve et al., 2014; Bliss et al., 2017) and active microwave backscatter from scatterometers (Drinkwater and Liu, 2000; Wang et al., 2011) and synthetic aperture radar (Mahmud et al., 2016; Howell et al., 2019). Despite the differences between the various data sources and methodologies, all of them have consistently revealed a statistically similar long-term trend of earlier SMO and delayed SFO (Markus et al., 2009; Bliss et al., 2017, Bliss and Anderson, 2018). In situ observations obtained during the Surface Heat Budget of the Arctic Ocean (SHEBA) experiment in the Beaufort Gyre region in 1997–1998 revealed that the SMO and SFO were primarily driven by pronounced atmospheric synoptic events, with specific dates triggered by a rain-on-snow event and a sequence of cold front passages, respectively. The SMO usually starts due to a large increase in downwelling longwave radiation and accompanied by moderate decreases in the surface albedo, and the SFO initiates after a step decrease of net surface energy flux (Persson, 2012). Reanalysis data also indicated that downwelling longwave radiation is the main factor in determining the variability of the SMO (Maksimovich and Vihma, 2012). And model results showed that positive anomalies of downward longwave radiation in spring and early summer initiated an earlier SMO (Kapsch et al., 2016).



The freeze-thaw cycle at the sea ice base is much different from that at the ice surface due to the additional regulation of the heat balance by the conductive heat flux through the ice column and the heat flux from ocean (Lei et al., 2018). In the Beaufort Gyre region, the amount of summer sea ice basal melt is generally comparable to or even larger than the surface melt (Perovich et al., 2015; Planck et al., 2020). The brine or fresh water injection associated with ice basal freezing and thawing processes is the main mechanism altering not only the physical hydrographic environment (Jackson et al., 2010; Randelhoff et al., 2017) but also the ecosystem (Ardyna and Arrigo, 2020; von Appen et al., 2021) of the underlying ocean. Despite their importance, the onset of basal ice melt and basal ice growth due to freezing (BMO and BFO) cannot be directly determined by any remotely-sensed radar or laser altimeter because of the nature of these methods (Ricker et al., 2017; Petty et al., 2020). Sea ice mass balance buoys (IMBs) on the other hand, which consist of a thermistor chain in combination with acoustic sounders above and below the ice, are able to provide seasonal sea ice mass balance observations from both the ice surface and ice base at a single point on a given ice floe (Perovich et al., 2021). Using such instruments, both surface and basal melt/freeze onsets as well as freeze-thaw cycles can be obtained at the measurement site along the ice Lagrangian drifting trajectory. Though IMBs are limited to one-dimensional ice mass balance measurements at some particular floes, the deployment site is usually chosen in an undeformed sea ice which is ideally representative for the ice conditions in a greater area (Planck et al., 2020). During the SHEBA campaign, IMB observations of undeformed ice at the Quebec 2 site in 1998 indicated that the surface melt was initiated by a rain-on-ice event on 29 May and ended by 17 August, while basal melt began in early June and ended in early October (Perovich et al., 2003). Planck et al. (2020) found that the BMOs at eight IMB sites in the Beaufort Gyre from 1997 to 2015 occurred within a relatively narrow window of 13 d in early June, and suggested some potential explanations such as warm water advection from the Bering Sea and ice basal energy budget. Based on measurements obtained by an "Ice-T" buoy deployed at the North Pole Environmental Observatory (NPEO) campaign in 2011, Vivier et al. (2016) found that the observed BMO in the Central Arctic Ocean preceded the SMO by 20 days. They ascribed this to an increased solar heating of the upper ocean through opening leads caused by storm events, highlighting the influence of synoptic events not only on sea ice surface freezing and thawing processes, but also that occurring at the ice bottom.

Another method to identify the sea ice freeze-thaw cycle is using the data measured by Upward Looking Sonars (ULS), which are usually deployed at the top of moorings in fixed geographic locations, measuring the submerged portion of the sea ice (ice draft). The ice draft can be converted to total sea ice thickness using an assumed ratio of ocean to ice densities, and also taking snow depth atop the ice into account (Krishfield et al., 2014). Analyzing the evolution of the probability distribution of the ice draft obtained from the ULS record in the Beaufort Gyre from 2003–2012, Krishfield et al. (2014) identified distinct seasonal cycles of sea ice thermodynamic growth and decay. Thus, the ULS measurements are a suitable tool for detecting the melt and freeze onsets defined by the thermodynamic processes (Smith and Jahn, 2019).



In essence, the basal melt and growth onset is controlled by the heat balance at ice-ocean interface, which is related to the thermodynamics of both sea ice and the upper ocean. During a number of field campaigns, Ice-Tethered Profilers (ITP) were co-located with IMBs to simultaneously monitor the thermodynamic processes related to the ice and the underlying ocean (Toole et al., 2011). It is convenient to study the influence of the upper ocean on the sea ice growth and decay based on the observation data. On a seasonal scale, the oceanic heat flux to the ice can be derived from two methods, i.e. by the sub-ice ocean water properties as measured by the ITP (Timmermans et al., 2011; Zhong et al., 2022), and by sea ice temperature and thickness changes derived from the IMB (Lei et al., 2018). Comparison of both measurements has so far show good agreement for the melting and freezing seasons in both the Arctic and Antarctic (Timmermans et al., 2011; Ackley et al., 2015).

In this study, we mainly focus on the characterization of the spatiotemporal variations in ice surface and basal melt and freeze onsets in Arctic Ocean by combining data of historical IMBs, passive microwave remote sensing, and ULS measurements. Further taking into account reanalysis data, we evaluate the surface radiation during the transition between freeze and thaw cycles. By co-analyzing IMB and ITP data, we also explore the connection of the basal melt and growth onset with heat fluxes from the surface and upper ocean. Based on our analysis, long-term variations in the patterns of the sea ice freeze-thaw cycle and their regional differences are revealed, and the coupling mechanisms between the sea ice melt-freeze cycle with the lower atmosphere and upper ocean are discussed.

## 2 Data and methods

### 2.1 Data

#### 2.1.1 Ice Mass balance Buoys

The main data source for this paper is the more than 100 Ice Mass balance Buoys (IMB) and Seasonal Ice Mass balance Buoys (SIMB) designed by the Cold Regions Research and Engineering Laboratory (CRREL) that have been deployed in the Arctic Ocean since 2000 (Perovich et al., 2021). In this paper, we refer to both types as IMB for simplicity. Each IMB is named by the year of deployment and followed by one letter in alphabetical order. The spatial coverage of the IMBs mainly extends into the Central Arctic Ocean (CAO, roughly located north of 80º N and with a bathymetry deeper than 1000 m) and the Beaufort Gyre (BG, roughly located between 70º and 80º N, 130º and 170º W with bathymetry deeper than 300 m). Some of the IMBs deployed on landfast ice are out of the scope of this study because shallow coastal waters have different ice-ocean coupling mechanisms, and are more vulnerable to terrigenous heat and freshwater inputs (Eicken et al., 2005). Their data is therefore not considered here. The IMB measures the distance of the respective acoustic sounder to the ice surface and ice base with an accuracy of 1 cm. Thus, both melt and freeze onsets of the ice surface and ice base at the deployment sites can be identified with high reliability. The surface melt and freeze onsets also can be related to observations of near-surface air temperature

collected by the IMB according to Rigor et al. (2000). Additionally, a thermistor chain with a vertical resolution of 0.1 m

provides temperature profiles through air, snow, ice, and ocean at an accuracy of 0.1°C, which can be used for an analysis of

the sea ice basal energy balance. In total, 69 IMBs are used in this study to detect the ice surface and/or basal melt and freeze

onsets in the BG and CAO for the period 2001–2018 (Figure 1).

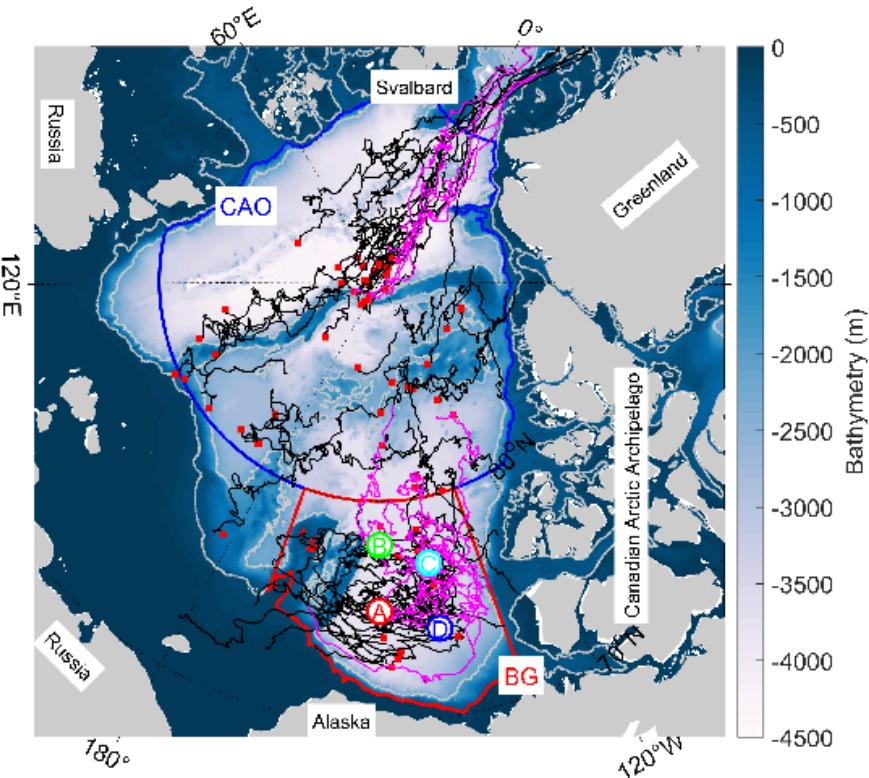

**Figure 1.** Deployment locations (red squares) and drift trajectories (solid lines) of 69 IMBs deployed in the Arctic Ocean in 2001−2018.
The pink lines are drifting trajectories of IMBs co-located with ITPs. The locations of four moored ULS in the Beaufort Gyre Observation
Systems are indicated by BGOS-A, B, C, and D. The Beaufort Gyre and the Central Arctic Ocean are defined by the areas with red and

blue boundaries, respectively. Colormap shows the bathymetry from 2-minute Gridded Global Relief Data (ETOPO2) v2 distributed by
NOAA National Centers for Environmental information (https://doi.org/10.7289/V5J1012Q). Grey contours donate the −300 and −2000 m
bathymetry.

### 2.1.2 Upward Looking Sonar data

Three or four upward looking sonars (ULS) were installed at the top of a number of moorings deployed beneath the BG sea

ice as parts of the Beaufort Gyre Observation System (BGOS-A, B, C, and D) every year since 2003, providing year-round

time series of ice draft (Proshutinsky et al., 2009). BGOS-A and BGOS-B were deployed along the 150º W meridian at 75º N

and 78º N, respectively. BGOS-C and BGOS-D were deployed along the 140º W meridian at 77º N and 74º N, respectively (Figure 1). After corrections for atmospheric pressure and speed of sound variations, the estimated error of the ice draft measurement is ± 0.05~0.10 m (BGOS ULS Data Processing Procedure, for details refer to Krishfield et al., 2014). Since daily

average of ice draft tends to be strongly affected by deformed ice resulting from dynamic growth and decay (Hansen et al., 2014), the daily median ice drafts are used to infer the thickness changes due to thermodynamic growth and decay, and subsequently also to identify the timing of the ice annual freeze-thaw cycle (Krishfield et al., 2014).

**2.1.3 Oceanographic data from Ice-Tethered Profilers**

Ice Tethered Profilers (ITP) designed by the Woods Hole Oceanographic Institution have been deployed in the Arctic Ocean

since 2004 to autonomously measure upper ocean properties at the depths between ~7 and 750 m (Krishfield et al., 2008a). The ITP measures seawater temperature, conductivity and pressure at a frequency of 1 Hz. Temperature and derived salinity are averaged into 2-db bins after application of a standard data processing procedure (Krishfield et al., 2008b). Some ITPs were co-deployed with IMBs, so that simultaneous measurements of seawater properties and sea ice basal freeze-thaw processes could be obtained. In this study, data measured by 12 ITPs (pink lines in Figure 1) are used when either the basal

melt or freeze onset are detected by the co-located IMB to investigate the coupling mechanism between the sea ice and the upper ocean. In addition, data from 17 ITPs deployed in the generally consistent area of central BG are used to characterize the decadal changes in spring sea surface temperature, which can indicate the changes in upper ocean contributed to enhanced sea ice melt. Here, we focus on a narrow region to eliminate spatial differences as much as possible.

**2.1.4 Ice surface melt and freeze onset from passive microwave data**

The satellite PMW dataset of surface melt and freeze onset dates are available from the NASA Cryosphere Science Research Portal, gridded to 25 km × 25 km (Markus et al., 2009; Stroeve et al., 2014). Based on the emissivity change due to the presence of liquid water, this dataset incorporates the PMW melt and freeze onset algorithm applied to passive microwave brightness temperatures collected over the period 1979–2020 from the Nimbus 7 Scanning Multichannel Microwave Radiometer (SMMR), the Special Sensor Microwave/Imager (SSM/I) and the Special Sensor Microwave Imager and Sounder (SSMIS).

The PMW dataset includes early melt and freeze onset dates, defined as the first day of ice surface melt or freeze, as well as continuous melt and freeze onset dates, defined as the day when ice surface melting or freezing conditions persist throughout the rest of the season. Here, we used these four records to identify the timing of ice surface melting or freezing at a given IMB location.

### 2.1.5 Sea ice concentration data

Daily sea ice concentrations along the IMB drift trajectories are derived from the Advanced Microwave Scanning Radiometer for EOS (AMSR-E) and its successor AMSR2 brightness temperatures (Meier et al., 2018) using the ARTIST Sea Ice (ASI) algorithm, with a spatial resolution of 6.25 km × 6.25 km (Spreen et al., 2008). To evaluate the impact of the absorption of short wave radiation by the ocean on sea ice freeze-thaw processes, a representative sea ice concentration around each buoy's location on a specific day is estimated by averaging concentration value of pixels within a radius of 50 km around the respective

buoy.

### 2.1.6 Atmospheric reanalysis data

The surface net shortwave and net longwave fluxes along the IMB trajectories are obtained from the European Centre for Medium-Range Weather Forecasts (ECMWF) ERA5 reanalysis dataset (https://cds.climate.copernicus.eu, Copernicus Climate Data Store, last access on 3 Apr 2022), which is produced using 4D-Var data assimilation and model forecasts in CY41R2 of

the ECMWF Integrated Forecast System. The ERA5 dataset extends from 1950 to 2020, with a horizontal spatial resolution of 0.25° × 0.25°, and a temporal resolution of 1 h. For evaluation of the surface atmospheric energy budget over the ice related to the ice freezing-thawing processes, the ERA5 data is daily averaged and bilinearly interpolated to a respective buoy's position.

### 2.2 Methods

#### 2.2.1 Detection of surface melt and freeze onsets

Three methods are applied for the detection of SMO and SFO. First, based on surface snow and ice mass balance observations and a combination of surface air temperature (SAT), SMO-IMB is defined as the date when the change of two subsequent daily records of surface position is negative, and the SAT is higher than $-1°C$. Correspondingly, the SFO-IMB is defined as the date of the when ice surface stops melting, and the SAT drops below $-1\ °C$ (i.e., from then on the ice surface is no longer melting).

Second, the ice surface melt and freeze onsets are detected using SAT, which has been a widely adopted method. Here, based on the SAT measured by the IMB, SMO-SAT and SFO-SAT are defined as the dates when observed daily SAT rises or drops below a threshold temperature of $-1°C$ after a 14-day running mean filter is applied (e.g., Rigor et al., 2000; Bliss and Anderson, 2018). Third, early melt onset (ESMO-PMW), continuous melt onset (CSMO-PMW), early freeze onset (ESFO-PMW), and continuous freeze onset (CSFO-PMW) of each buoy location are derived from PMW satellite observation (Markus et al., 2009)

along the respective buoy trajectory. However, the PMW data is not available in the vicinity of the North Pole due to the constrained satellite orbit.





### 2.2.2 Detection of basal melt and growth onsets using IMB and ULS data

The basal melt (BMO-IMB) and growth onset (BFO-IMB) are identified from IMB observations as the date when the ice bottom elevation reaches the lowest (largest basal ice growth) or highest (largest basal ice melting) positions, respectively, after applying a 14-day running mean filter. The potential interference in BFO-IMB detection caused by false ice bottom formation (Eicken, 1994) in the melt season is carefully identified and excluded. The IMB observation typically showed a sufficiently basal growth and following by a rapidly thinning in early to mid-summer without any significant atmospheric and oceanic temperature signals if false bottom existed (Smith, et al, 2022). In this case, BFO-IMB is detected after the false bottom formation. A second set of indices indicating ice basal melt and growth onset are derived from ULS daily median ice draft data. The ULS measures the total ice draft, which is usually integrating both ice surface and basal melt and freeze processes. Thus, we cannot separate the changes in ice surface and bottom using ULS data. However, we can obtain a total 1-D ice volume tendency from this dataset. First, the climatological ice thickness at each mooring site is derived to remove the irregularly fluctuating data from the time series to separate the modal ice from the ridged ice. Then, the MO-ULS and FO-ULS are defined as the date when the smoothed ice thickness reaches the maximum or minimum value after applying a 30-day running mean filter. In the case when sea ice has vanished completely in summer, FO-ULS is defined as the first day when a persistent ice cover is continuously observed.

**Table.1** Definition of different surface and basal melt and freeze onsets

| Variables | Definition |
| --- | --- |
| SMO-IMB/SFO-IMB | surface melt and freeze onset based on surface elevation from IMBs |
| SMO-SAT/SFO-SAT | surface melt and freeze onset based on surface air temperature from IMBs |
| SMO-EPMW/SFO-EPMW | early surface melt and freeze onset from PMW along the respective buoy trajectory |
| SMO-CPMW/SFO-CPMW | continuous surface melt and freeze onset from PMW along the respective buoy trajectory |
| BMO-IMB/BFO-IMB | basal melt and growth onset based on basal elevation from IMBs |
| MO-ULS/FO-ULS | melt and growth onset based on sea ice draft from ULS |

### 2.2.3 Estimation of conductive heat flux and oceanic heat flux at the ice-ocean interface

To avoid the high porosity skeletal layer near the ice base (Lei et al., 2014), the bulk conductive heat flux is investigated for a



specified reference layer defined at 0.2-0.6 m above the ice base, and estimated by

$$F_c = k_i \frac{\partial T_i}{\partial z},$$ (1)

where $k_i$ is the sea ice thermal conductivity and $\partial T_i/\partial z$ is the vertical ice temperature gradient. $k_i$ is a function of sea ice

temperature and salinity (Untersteiner, 1961). According to McPhee (1992) and McPhee et al. (2003), the oceanic heat flux

from the mixed layer into the sea ice primarily depends on the amount of surface mixed layer heat, which is characterized by

ocean mixed layer temperature departure from the freezing point ($\Delta T$), as well as on the turbulent mixing in the boundary layer,

characterized by the friction speed, $\boldsymbol{u_{*0}}$. Operationally, $\Delta T$ is calculated using the topmost valid data from an ITP dataset, if

that depth is shallower than 20 m. $\boldsymbol{u_{*0}}$ is calculated as

$$\frac{\kappa \boldsymbol{V}}{\boldsymbol{u_{*0}}} = log \frac{|\boldsymbol{u_{*0}}|}{f z_0} - A - iB,$$ (2)

where $V$ is the difference between ice velocity and surface geostrophic current velocity, $f$ is the Coriolis parameter, $z_0$ is the

hydraulic roughness of the ice bottom with a typical value of 0.01 m for undeformed multiyear sea ice, and $A$ and $B$ are

constants with values of 2.12 and 1.91, respectively (McPhee et al., 2003). The geostrophic current velocity is relatively small

in the Arctic pack ice zone, typically less than 5 cm s⁻¹, which can be neglected (Krishfield and Perovich, 2005). Then, the

oceanic heat flux $F_w$ is estimated as:

$$F_w = \rho_{sw} c_p C_H u_{*0} \Delta T,$$ (3)

where $\rho_{sw}$ and $c_p$ are the density and specific heat of seawater, respectively, and $C_H = 0.006$ is a bulk heat transfer coefficient

(McPhee, 1992).

## 3 Results and discussions

For each IMB trajectory, four pairs of surface melt and freeze onsets and one pair of basal melt and growth onsets are derived

(Table S1). For example, IMB 2013F was operational for more than 700 d, from 25 August 2013 to 27 August 2015, covering

two full ice growth seasons and one full ice melt season (Figure 2). Following the methods outlined above, the SMO and SFO

from IMB, SAT, and PMW along the buoy's trajectory are identified. ESMO-PMW on 02 May 2014 was triggered by a spring

storm event and was about one and a half month earlier than the SMO-SAT, CSMO-PMW and SMO-IMB. Apart from that,

CSMO-PMW, SMO-SAT and SMO-IMB dovetail nicely, followed upon by a rapid decrease of snow depth. All 2014 SFOs

derived from the different methods were highly consistent as well. Both the remotely-sensed and in situ surface air temperature



measurements captured the surface snow accumulation processes similarly well. For the 2015 SMOs, ESMO-PMW (19 May

2015) captured the surface snow melt onset well, giving the same result as SMO-IMB. The SMO-SAT (11 June 2015) occurred

22 d later. Compared to the surface melt, the 2014 BMO-IMB (8 May 2014) occurred more than one month earlier than the

SMO-IMB (13 Jun 2014) and was very close to the ESMO-PMW, which might have been caused by the enhanced solar

radiation deposited and increased ocean mixing during spring storms (e.g., Viver et al., 2016). In contrast, the 2015 BMO-IMB

(26 May 2015) was 7 d later than the SMO-IMB. The 2014 BFO-IMB (5 Oct 2014) was approximately one month later than

the SFO-IMB (2 Sep 2014). Following this example, the differences of ice surface melt and freeze onsets among the various

methods and of the melt and freeze onsets between the ice surface and base are then investigated for all available datasets.

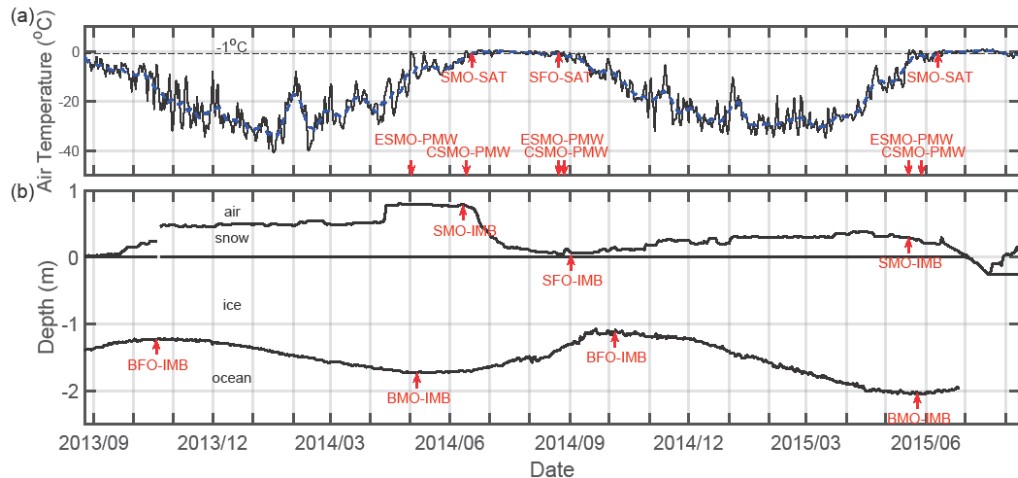

**Figure 2.** Detection of surface and basal melt and freeze onsets from IMB 2013F: (a) daily near-surface air temperature (black solid line)
and its 14-day running mean (blue dashed line); (b) snow depth and sea ice thickness, with the zero-line denoting the initial snow/ice interface.
The arrows indicate the melt and freeze onset dates estimated by the different methods.

### 3.1 Comparison of ice surface melt and freeze onsets from different methods

The four different SMOs and SFOs of 55 IMB trajectories are intercompared to each other. For the surface melt onset, SMO-

SAT and CSMO-PMW matched best with the smallest deviation of less than 2 d (Fig. 3), which means that the SAT threshold

method captured the process of continuous ice surface melt quite reliably. The SMO-IMB was about 8~9 d earlier than SMO-

SAT and CSMO-PMW, and 4 d later than the ESMO-PMW. Similar to IMB2013F (Figure 2), the moderate deviations between

SMO-IMB and SMO-SAT were mainly caused by spring storm events. Warm moisture carried by synoptic events from lower

latitudes could lead to the SAT reaching the threshold temperature in a transitory period and promote surface snow melt.

However, such a temperature impulse could be missed by the 14-day running mean filter. After the spring storms, the observed

SAT dropped down, and then increased and retained above the threshold temperature until the commencing of continuous



surface melt.

For the surface freezing onset, both the SFO-IMB and SFO-SAT matched well with ESFO-PMW, with 2 d later and 3 d earlier than ESFO-PMW, respectively. And SFO-IMB occurred 8 d later than the SFO-SAT, which could also be attributed to the synoptic events and running mean filter just as surface melt onset. Autumn storms brought about several temporary freeze-thaw cycles (i.e. snow fall and surface melting) before fully freezing ensured. Thus, the transition date when filtered SAT dropped below the threshold temperature was earlier than the date when surface melt terminated. CSFO-PMW were always later than others, with an average delay of 13 d after the SAT decreases below freezing.

In general, although the surface freeze-thaw cycles detected by the three methods show some moderate deviations, both surface melt and freeze onsets from the PMW and SAT methods generally match the results from the IMB observation quite well. In particular, the SMO-SAT and SFO-SAT reliably capture the "inner" melt season, defined as the period between the CSMO-PMW and the ESFO-PMW (Markus et al., 2009). Here, the SMO-SAT and SFO-SAT are hereinafter used as the general SMO and SFO for the purposes of comparing to the BMO and BFO and of identification of the spatiotemporal variation, because the SMO-IMB and SFO-IMB were not available at some buoy sites.

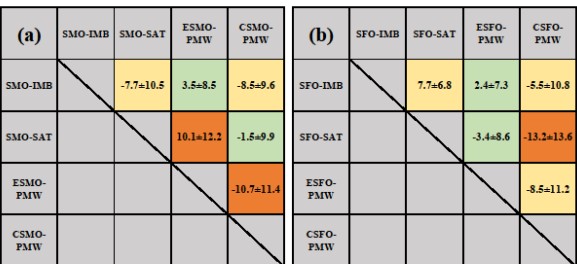

**Figure 3.** Differences (in days) of surface melt onset (SMO) (a) and surface freeze onset (SFO) (b) determined by the various methods. The small (< 5 d), median (5–10 d) and large (> 10 d) deviations are indicated by the color of green, yellow and orange, respectively. Unit: days.

### 3.2 Temporal and spatial variations of ice surface and basal melt and freeze onsets

The SMO, SFO, BMO, and BFO derived from 69 IMBs in the BG and CAO are presented in Figure 4. The SMO ranged from mid-May to early July, with a mean on 11 June (± 9 d). The BMO ranged from late May to early July, with a mean on 5 June (± 8 d), approximately 6 d earlier than the SMO. The SFO ranged from early August to early September, with a mean on 20 August (± 8 d). The BFO ranged from early October to late December, with a mean on 14 November (± 21 d). The average BFO lagged behind the corresponding SFO by almost three months. As a result, the average basal melt season was nearly three months longer than that at surface, which was dominated by later onset of basal ice growth.



The spatial variations in surface and bottom melt and freeze onsets were also quite remarkable (Figure 4). The overall spatial patterns revealed a decreasing trend of surface and basal melt onset and a corresponding increasing trend of freeze onsets with an increase of latitude, as one would expect. Sea ice generally melts earlier and freezes later in the BG compared to the CAO. The trends of SMO and BMO against the latitude were $0.6 \pm 0.2$ d degree$^{-1}$ ($p<0.001$) and $2.0 \pm 0.2$ d degree$^{-1}$ ($p<0.001$), respectively (Fig. 4e). In the BG, the average BMO (23 May) occurred approximately 17 d earlier than the SMO

(9 June) for the ice with a thickness of $2.36 \pm 0.76$ m. In the CAO, the BMO and SMO usually occurred almost at the same time (18 June vs 14 June) for the ice with a thickness of $2.23 \pm 0.47$ m. The trend of the SFO against the latitude was $-1.0 \pm 0.3$ d degree$^{-1}$ ($p<0.001$), while the BFO exhibited a considerable amount of scatter with increasing latitude (Fig. 4e). The relevant mechanisms will be discussed later.

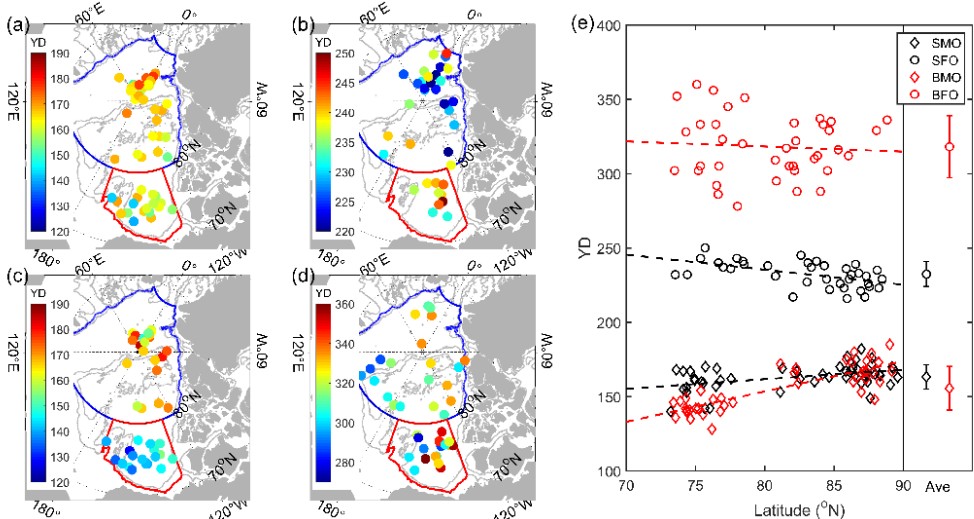

**Figure 4.** Timing of ice surface and bottom melt and freeze onsets of all sites: (a) SMO, (b) SFO, (c) BMO, and (d) BFO. The color codes (note the different scales for different panels) indicate the respective dates (day of the year). Grey contours denote the 300 and 2000 m isobaths. (e) Variations in dates of melt and freeze onset as a function of the latitude.

**3.3 Surface radiation budget during the transition between ice surface melting and freezing**

Here, we extract the topmost snow temperatures from IMB temperature profiles to determine the thermodynamic state of snow

during the transition period from freezing to melting. Since the vertical resolution of the temperature profiles is only 0.1 m, the data of 33 IMBs with snow depths larger than 0.1 m are used. The topmost snow temperatures were averaged over 10 d before and after melt onset for each IMB. The average topmost snow temperature increased from $-1.6 \pm 1.2$ °C for the 10 days before the SMO to $0.1 \pm 0.5$ °C for the 10 days after the SMO. Thus, the increase in topmost snow temperature crossing the melting point can be considered as one of the preconditions of the SMO.



Similarly, using ERA5 reanalysis data, we evaluate the average net shortwave and net longwave radiation fluxes over a
period of 10 d before and after the surface melt and freeze onsets for all available IMBs, with a positive value denoting a
downwelling heat flux. The average net shortwave radiation was $81.1 \pm 18.3$ W m$^{-2}$ and $84.9 \pm 17.4$ W m$^{-2}$ for the 10 days
before and after the SMO respectively, which amounts to an increase of 3.8 W m$^{-2}$. The corresponding average net longwave
radiation was $-38.2 \pm 10.1$ W m$^{-2}$ and $-26.9 \pm 10.1$ W m$^{-2}$ for the 10 days before and after the SMO, which amounts to an

increase of 11.3 W m$^{-2}$ (or the net loss decreased). These results are in line with previous findings stating that the ice surface
melt onset in the Arctic is primarily triggered by an increase of longwave radiation (Maksimovich and Vihma, 2012; Persson,
2012). Warm and moist air masses carried northwards from lower latitudes by distinct synoptic events would increase the
downwelling longwave radiation as well as the net longwave radiation, while the net shortwave radiation wouldn't be altered
too much due to the high snow albedo at the onset of surface melt (Persson, 2012). For the surface freeze onset, the average

net shortwave radiation was $52.8 \pm 16.1$ W m$^{-2}$ and $38.4 \pm 14.7$ W m$^{-2}$ for the 10 days before and after the SFO, respectively,
which amounts to a decrease of $-14.4$ W m$^{-2}$. Similarly, the average net longwave radiation was $-15.5 \pm 6.6$ W m$^{-2}$ and $-20.2$
$\pm 7.1$ W m$^{-2}$ for the 10 days before and after the SFO, which amounts to a decrease of $-4.7$ W m$^{-2}$. These contrasting results to
the melt onset conditions are expected, as the SFO is primarily controlled by the decline of net shortwave radiation as the
approaching of polar night.

**Table 2.** Average surface net radiation changes during the transition from surface melting to freezing.

|  | SMO | | | SFO | | |
| --- | --- | --- | --- | --- | --- | --- |
|  | SMO-10d | SMO+10d | Difference | SFO-10d | SFO+10d | Difference |
| Shortwave radiation (W m$^{-2}$) | $81.1 \pm 18.3$ | $84.9 \pm 17.4$ | 3.8 | $52.8 \pm 16.1$ | $38.4 \pm 14.7$ | $-14.4$ |
| Longwave radiation (W m$^{-2}$) | $-38.2 \pm 10.1$ | $-26.9 \pm 10.1$ | 11.3 | $-15.5 \pm 6.6$ | $-20.2 \pm 7.1$ | $-4.7$ |

### 3.4 Heat balance at the ice-ocean interface during basal melt and growth onsets

        Here, we compare $\Delta T$, $u_{*0}$, $F_w$, and $F_c$ during the 10 days before (-10 d) and after (+10 d) the calculated BMO using data
of 12 ITPs (Table 3). The average $\Delta T$ ($-10$ d) ranged from 13 to 45 mK, with a mean of $28 \pm 10$ mK. This was warmer than
the typical winter mixed layer temperature (~ several mK) (Shaw et al., 2009). The average $\Delta T$ ($+10$ d) was almost twice as

large, with a mean of $56 \pm 22$ mK. $u_{*0}$ did not show any significant changes. Therefore, the change of ocean heat flux to the
ice should be governed by the thermodynamic processes rather than the dynamics. The estimated $F_w$ increased from $3.0 \pm 1.2$
to $6.8 \pm 2.7$ W m$^{-2}$, which was substantially larger than the typical winter value of about $1.0 \pm 2.9$ W m$^{-2}$ in the Canada Basin
(Cole et al., 2014) and $2.1 \pm 2.3$ W m$^{-2}$ in the Eurasian Basin (Peterson et al., 2017). During the cold winter, the typical sea ice
temperature profile is almost linear (Lei et al., 2014). As the summer approaches, the upper ice column warms faster than its





lower portion, leading to a "C-shape" vertical temperature profile with a gradually reduced temperature gradient at the ice base (Lei et al., 2014). Correspondingly, the upward average conductive heat flux at the ice bottom decreased from $4.4 \pm 1.5$ W m$^{-2}$ to $3.2 \pm 1.7$ W m$^{-2}$. The key to BMO is when $F_w$ becomes greater than $F_c$. Therefore, once the upward oceanic heat flux surpasses the upward conductive heat flux at the ice bottom, ice basal melt commences.

The surface ocean typically warms earlier in the BG compared to the CAO due to the larger amount of incoming solar

radiation in lower latitudes. At the same time, both remote sensing and models revealed that sea ice in the BG shows higher divergence and larger water fraction compared to the CAO (Wernecke and Kaleschke, 2015, Wang et al., 2016), as well as a higher fraction of thin ice (Petty et al., 2020). Consequently, the upper ocean in the BG absorb more solar radiation compared to the CAO (e.g., Perovich et al., 2011). This may at least partly explain why in the BG the BMO occurred much earlier than the SMO, while they occurred almost at the same time in the CAO.

**Table 3.** Summary of the changes of oceanic heat flux from the observations of ITP on BMO.

| IMB | ITP | Location | BMO (yyyy/mm/dd) | $F_c$ (W m$^{-2}$), (−10 d) | $F_c$ (W m$^{-2}$), (+10 d) | $\Delta T$ (m K), (−10 d) | $\Delta T$ (m K), (+10 d) | $u_{*0}$ (cm s$^{-1}$), (−10 d) | $u_{*0}$ (cm s$^{-1}$), (+10 d) | $F_w$ (W m$^{-2}$), (−10 d) | $F_w$ (W m$^{-2}$), (+10 d) |
|---|---|---|---|---|---|---|---|---|---|---|---|
| 2005B | ITP3 | BG | 2006/06/03 | 3.7 | 2.6 | 29 | 51 | 0.51 | 0.45 | 3.6 | 5.6 |
| 2006C | ITP6 | BG | 2007/05/28 | 6.0 | 5.3 | 45 | 56 | 0.36 | 0.49 | 4.0 | 6.7 |
| 2006C | ITP6 | CAO | 2008/06/07 | 4.4 | 3.2 | 36 | 79 | 0.45 | 0.54 | 4.0 | 10.4 |
| 2007D | ITP7 | CAO | 2007/06/23 | 1.8 | -0.7 | 17 | 33 | 0.34 | 0.40 | 1.4 | 3.3 |
| 2007E | ITP18 | BG | 2008/05/20 | 6.4 | 5.1 | 39 | 73 | 0.49 | 0.40 | 4.7 | 7.2 |
| 2007F | ITP13 | BG | 2008/05/28 | 5.9 | 4.4 | 31 | 85 | 0.47 | 0.58 | 3.5 | 11.3 |
| 2007J | ITP11 | CAO | 2008/06/20 | 2.7 | 1.9 | 31 | 56 | 0.24 | 0.37 | 1.9 | 5.1 |
| 2007J | ITP11 | BG | 2009/05/28 | 5.9 | 5.0 | 21 | 89 | 0.25 | 0.48 | 1.3 | 10.5 |
| 2008E | ITP19 | CAO | 2008/06/08 | 4.5 | 3.3 | 26 | 32 | 0.40 | 0.93 | 2.5 | 7.3 |
| 2010A | ITP38 | CAO | 2010/06/16 | 1.9 | 1.1 | 13 | 23 | 0.44 | 0.61 | 1.5 | 3.5 |
| 2012L | ITP65 | BG | 2013/05/21 | 4.8 | 4.1 | 21 | 32 | 0.91 | 0.63 | 4.7 | 4.9 |
| 2013B | ITP61 | CAO | 2016/06/09 | 3.8 | 2.5 | 20 | 44 | 0.62 | 0.85 | 3.0 | 9.1 |
| 2014I | ITP85 | BG | 2015/05/24 | 5.0 | 3.4 | 45 | 77 | 0.48 | 0.60 | 2.5 | 6.6 |
| 2015D | ITP83 | CAO | 2015/06/16 | 4.3 | 3.0 | 20 | 48 | 0.69 | 0.28 | 3.3 | 3.3 |
| Average | | | | 4.4±1.5 | 3.2±1.7 | 28±10 | 56±22 | 0.48±0.18 | 0.54±0.18 | 3.0±1.2 | 6.8±2.7 |



Here we further investigate the mechanism relevant to the BFO from the perspectives of both sea ice itself and underlying ocean. According to the heat balance at the ice-ocean interface, sea ice basal growth begins when the heat transfers away from the ice bottom due to the upward conductive heat flux surpasses the heat transfer into the ice by the oceanic heat flux. As shown in the IMB 2007J data (Figure 5b), the relatively warm sea ice column in summer began to cool downward from the surface to the bottom as a result of the decreasing SAT. Correspondingly, the conductive heat flux in the sea ice basal layer gradually shifted from downward to upward as the freezing front gradually moved downward through the ice column. At the same time, data from the co-located ITP11 revealed that the remaining heat stored in the mixed layer is transported upwards to continue to melt the ice base or delay ice growth. Subsequently, the mixed layer temperature decreased gradually to the freezing point (Figure 5c), which took ~63 d since the SFO. During this period, the ice floe drifted southward into a region with much warmer subsurface water just beneath the mixed layer, i.e. a stronger near-surface temperature maximum (NSTM, Figure 5d). As a result, although the oceanic mixed layer temperature dropped to the freezing point in late October 2008, basal freezing did not commence until mid-November 2008, when the upward conductive heat flux increased to $> 10$ W m$^{-2}$. The upward heat transport was balanced by the subsurface oceanic heat release from the NSTM (Figure 5c), which delayed basal ice growth by approximately 21 d. The heat stored in the NSTM was held in place by the strong stratification of the summer halocline, and was finally released by a halocline erosion induced by shear-driven entrainment (Toole et al., 2010; Jackson et al., 2012, Lin and Zhao, 2019). In summary, the propagation of the freezing front from the ice surface to the bottom, combined with the oceanic heat release from the mixed layer and subsurface ocean, jointly generated the time delay of approximately 3 months between the BFO and SFO.

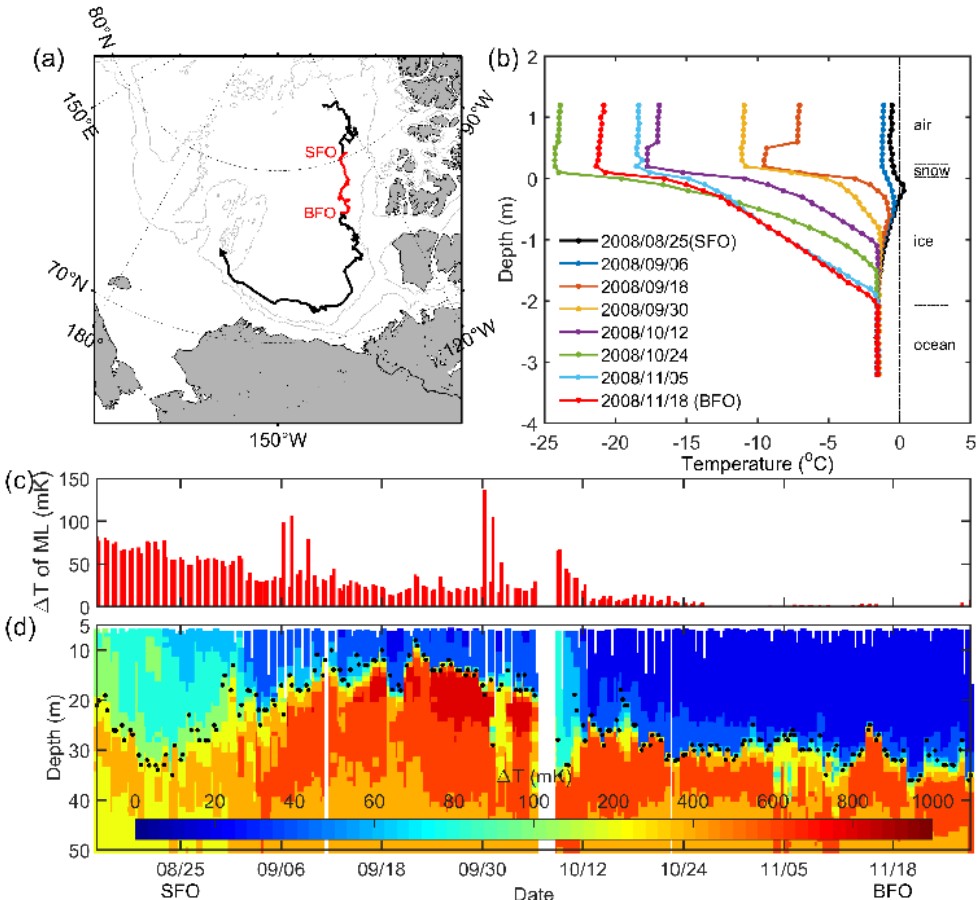

**Figure 5.** (a) Drift trajectory of co-located IMB 2007J and ITP 11; (b) IMB 2007 temperature profiles from SFO (black line) to BFO (red line); (c) mixed layer temperature deviation from the in situ freezing point; (d) ITP 11 profiles of temperature deviation from the in situ freezing point. Black dots donate the mixed layer depth where the potential density relative to 0 dbar first exceed the shallowest sampled density by 0.01 kg m⁻³. The white vertical bar in early October 2008 indicates a data gap.

**3.5 Impact of air temperature and ice thickness on onset of basal ice growth**

As explained above, the cooling of the sea ice is one of the preconditions of basal ice growth. Thus, we quantitatively analyze the role of sea ice cooling on the BFO here. In order to minimize the impact of spatial variations, we use the time delay between BFO and SFO instead. Near-surface air temperature, snow depth, ice thickness and ice internal structure (brine volume fraction) are suggested as the crucial factors controlling the cooling efficiency of the sea ice column, or worded differently, and the propagation efficiency of the freezing front from the sea ice surface to the bottom (which is the prerequisite for the BFO). The influence of ice internal structure cannot be assessed using IMB data, thus we don't consider its influence here. The lower SAT accelerates the sea ice cooling, while a thicker snow cover plays the opposite role of a thermal insulator due to its low thermal



conductivity. Thereby, an ice cooling index (*ICI*) is introduced here as

$$ICI = H_{is}/FDD, \tag{4}$$

where, $H_{is} = H_i + k_i/k_s \cdot H_s$. $\tag{5}$

The $H_{is}$, $H_i$ and $H_s$ are the equivalent ice thickness, ice thickness and snow depth, respectively, $k_i$ (2 W m⁻¹ K⁻¹) and $k_s$ (~ 0.3 W m⁻¹ K⁻¹) are the thermal conductivities of ice (Yen et al., 1991) and snow (Sturm et al., 2002), respectively. *FDD* is the amount of seasonal cumulative freezing degree days, which is defined as the time-integrated daily air temperature below the seawater freezing point (–1.8 °C), with the SFO as the zero reference. $H_i$ is defined as the mean ice thickness between the SFO and BFO because the ice thickness at the BFO can be thinner by as much as > 0.50 m compared to that at the SFO. The IMB

data showed that snow accumulation usually occurred just after the SFO, with snow mostly accumulating in the early freezing season and maintaining a steady state until the surface melt occurred. Thus, $H_s$ is defined as the mean snow depth between the SFO and BFO. The time delay between the BFO and SFO ranged from 38 to 115 d, with a mean of 82±26 d. For consistency and comparability, we defined the same period for *FDD* integration starting from SFO. The most significant relationship between the *ICI* and the time lag between the BFO and SFO was found if the integration time is chosen as 45 days, with

$R^2=0.81$, $p<0.01$ (Figure 5).

Generally, the lower surface air temperature, thinner sea ice and snow cover are related to the earlier basal ice growth, and vice versa, suggesting the time lag between the BFO and SFO can be significantly attributed to the ice column cooling efficiency. Without considering the SAT, $H_{is}$ also has a significant relationship with the time lag ($R^2=0.79$, $p<0.05$) because the SAT does not show significant differences between the buoys. These results imply a negative feedback, i.e., thinner snow and

ice favor earlier basal freeze-up in the following winter. Since sea ice thickness and snow depth of each IMB vary in a wide range, that is the most likely explanation why the BFO exhibits a much larger variability.

**Table 4.** Ice cooling index (*ICI*) and time lag between BFO and SFO

| IMB | $H_i$(m) | $H_s$(m) | *FDD*(K.d) | *ICI*(m K.d⁻¹) | $\Delta H_i$(m) | BFO-SFO (d) |
|---|---|---|---|---|---|---|
| 2002A | 2.13 | 0.30 | 190 | 0.0217 | 0.16 | 74 |
| 2004A | 1.90 | 0.44 | 267 | 0.0181 | 0.17 | 97 |
| 2004B | 1.86 | 0.35 | 97 | 0.0433 | 0.07 | 103 |
| 2004D | 1.66 | 0.36 | 122 | 0.0335 | 0.24 | 109 |
| 2011J | 2.41 | 0.28 | 108 | 0.0400 | 0.15 | 115 |
| 2011K | 0.64 | 0.15 | 188 | 0.0088 | 0.63 | 52 |





| | | | | | |
|---|---|---|---|---|---|
| 2012I | 0.80 | 0.28 | 118 | 0.0223 | 0.27 | 77 |
| 2013F | 0.66 | 0.29 | 170 | 0.0152 | 0.20 | 38 |
| 2015F | 1.25 | 0.14 | 131 | 0.0164 | 0.25 | 74 |

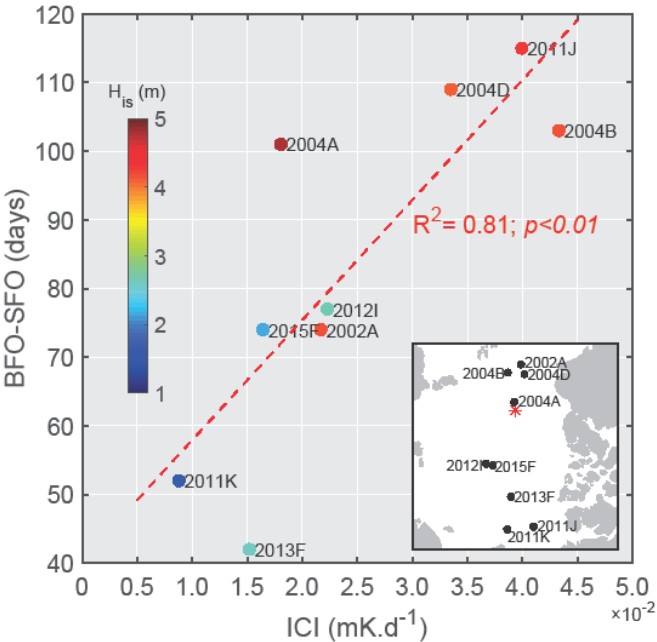

**Figure 6.** The relation between ice cooling index (*ICI*) and the time lag between the onset of surface freeze and basal ice growth. Colormap represents the equivalent ice thickness.

**3.6 Relationship between the melt and freeze season length and the amount of ice melt/freezing**

Sea ice surface melt includes surface snow melt and surface ice melt. The equivalent surface melt (*ESM*) is defined as:

$$ESM = (\rho_s/\rho_i)H_s + H_i, \tag{6}$$

where $H_s$ and $H_i$ are surface snow and ice melt, respectively, $\rho_s = 300$ kg m$^{-3}$ is the snow density and $\rho_i = 900$ kg m$^{-3}$ is the ice density (Perovich et al., 2014). Here, surface snow and ice melt are inferred from a combination of surface acoustic sounder observations and the distinct difference of temperature gradients in air, snow and ice. In agreement with a previous study by Lindsay (1998), the total surface melt was closely correlated with the length of the surface melt season ($R^2$=0.48, $p < 0.01$), and increasing by 0.07 m with a lengthening of the surface melt season by 10 d (Figure 7a). Surface melt is determined by the surface energy balance, which is influenced by surface albedo, radiation and turbulent fluxes, as well as wind erosion and evaporation (Persson, 2012). On the basis of SHEBA data, Perovich et al. (2003) found that a thinner snow cover was related



to an earlier surface ice melt, but that the initial snow depth seems to have had no impact on the total surface melt. Here, based on the IMB observations, the initial snow depth at the surface melt onset exhibits a close correlation with the total surface melt ($R^2 = -0.52$, $p < 0.01$; Figure 7b), which is a manifestation of the well-known albedo feedback. The reason for the above differences may be that the ice mass balance observations were carried out on a variety of different ice surface topographies

during SHEBA, which were likely susceptible to surface melt water redistribution and horizontal heat advection, while the IMB deployment locations are usually biased towards level ice.

As shown in Figure 7c, no direct relationship was identified between the total basal melt and the basal melt season length. Larger basal melt was always accompanied by a longer basal melt season, but the opposite was not always true. Actually, the ice concentration conditions around the IMB buoy can affect the amount of ice basal melt by adjusting the shortwave radiation

budget of ice-ocean system. For comparison, the lengths of the basal melt season of IMBs 2004A, 2005B and 2006C were 170, 174 and 172 d, respectively. However, the mean June-September ice concentration along the drifting trajectories of IMB 2004A, 2005B and 2006C were distinctly different, with values of 99%, 89%, and 71%, respectively. As a result, the basal melt at IMB 2006C (2.14 m) was nearly three times larger than at IMB 2005B (0.77 m) and ten times larger than at IMB 2004A (0.22 m) because the lower ice concentration causes more shortwave radiation to be absorbed by the upper ocean. This suggests that the

total basal melt does not significantly correlate with the initial ice thickness (Figure 7d), but is more likely related to the amount of solar heat input into the upper ocean in summer. The relatively high ice concentration at IMB 2004A also can explain why there was a larger time lag between the onset of surface ice freeze and basal ice growth relative to the linear regression as shown in Figure 6. If we exclude the individual BMO impacted by early spring storms, such as BMO of IMB 2013F in 2014, the BMO was significantly correlated with the total basal melt ($R^2=0.82$, $p < 0.01$), i.e., earlier BMO always lead to more basal

melt (not shown).

It is also notable that the total basal growth shows a significant correlation ($R^2=0.63$, $p < 0.01$) with the length of the basal freeze season (Figure 7e). As investigated above, basal growth of thinner sea ice started earlier compared to thicker ice. In combination with the negative conductive feedback, i.e., thinner ice grows faster than thicker ice, thinner ice generally experienced a longer freezing season and a larger ice growth. Considering the thermal insulation effect of the snow cover, the

initial equivalent ice thickness $H_{is}$ (defined above) was used to identify the link between the initial ice thickness and the total ice growth. As shown in Figure 7f, the total sea ice growth during the entire freezing season increased by 0.26 m with the initial $H_{is}$ decreasing by 1 m. For all IMBs that experienced the complete melting or freezing seasons, the average ice melt was 0.56 m at the surface and 0.65 m at the ice bottom, while the average ice growth was 0.74 m. The average annual ice thickness budget derived from all IMB observations during 2000-2018 amounts to –0.47 m, which clearly shows the ongoing reduction

of the Arctic sea ice thickness. This finding is also consistent with the February/March ice thickness retrieved from the satellite





altimeter measurement of ICESat, which indicated a decrease of ~0.37 m or ~20% thinning across an inner Arctic Ocean domain from 2008 to 2019 (Petty et al., 2020). We infer that the growth of sea ice in winter is not sufficient to compensate for the melt in summer, even though the negative conductive feedback enhances the ice growth during the freezing season.

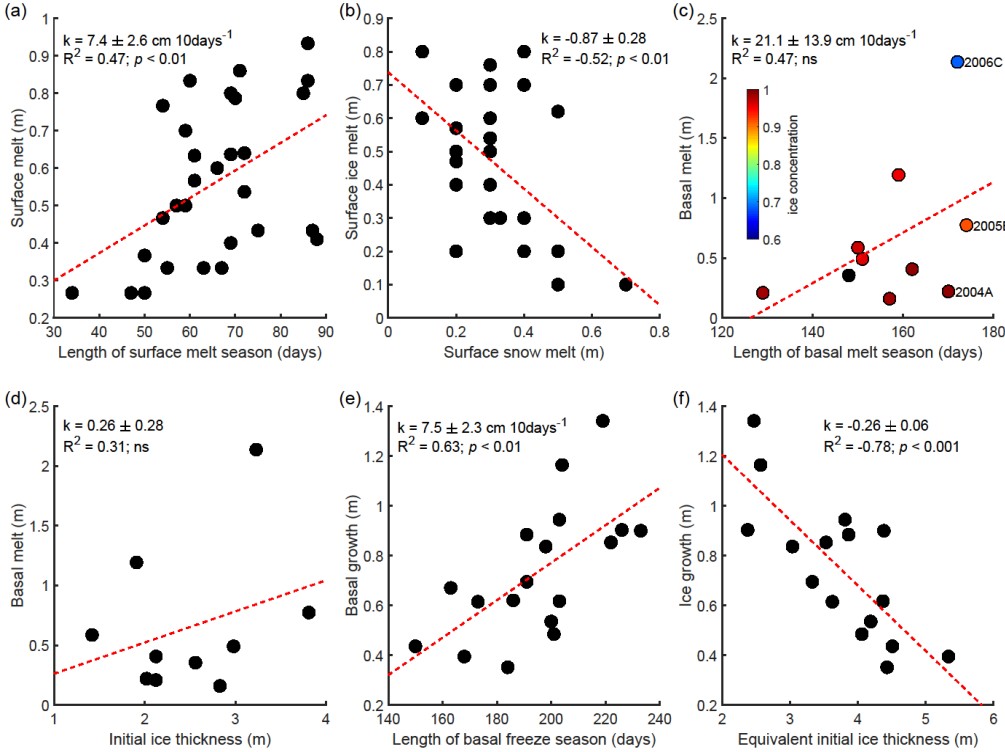

**Figure 7.** Relationships between (a) length of surface melt season and total surface melt; (b) surface snow melt and surface ice melt; (c) length of basal melt season and total basal melt; (d) initial ice thickness and basal growth; (e) length of basal freeze season and total basal growth; (f) equivalent initial ice thickness and total ice growth.

### 3.7 Decadal changes of basal melt and freeze season length in the Beaufort Gyre

As shown in Figure 3c and 3d, the basal melt and freeze onsets derived from IMB observations revealed a large spatial variation, especially in the CAO. In order to minimize the impact of spatial variations, we estimate here the decadal changes in the BMO and BFO in the BG using a synthetic analysis (Figure 8). All the BMOs and BFOs detected in the IMB observations were equally divided into two 9-year periods. The average BMOs were on 31 May in 2001–2009 (9 cases), and 24 May in 2010–2018 (17 cases), respectively. Similar to the SMO, there is also an earlier trend of the BMO, which occurred approximately 7 d earlier in the recent 9 years compared to the previous 9 years. Although the trend was relatively weak, the ITP observations in the central BG indicate that the average oceanic mixed layer temperature departure from the local freezing point (*ΔT*) in May was 24.1 mK in the recent 9 years and 21.5 mK in the previous 9 years (Figure 9), which can be partly explained by more

frequent lead openings in early spring (e.g., Qu et al., 2021). Therefore, there is a positive feedback, that thinner and more

vulnerable ice results in earlier BMO, and thus makes ice thinner and more vulnerable. However, the average BFOs were 15

November in 2010–2018 (15 cases), which was 8 d earlier than (23 November) in 2001–2009 (9 cases), which can be related

to thinning ice thickness  (1.30 m vs 1.83 m).

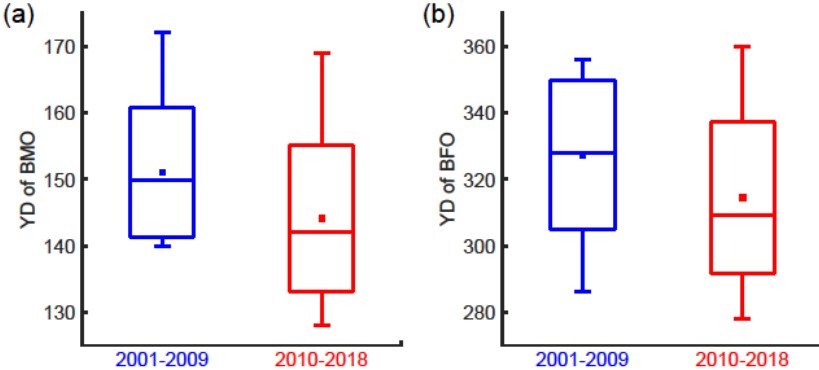

**Figure 8.** Decadal changes in BMO (a) and BFO (b) obtained from Lagrangian IMB observations in the Beaufort Gyre. The solid square is
the mean, the horizontal line is the median, the box represents ±1 standard deviation and the whiskers are the maximum and minimum values.

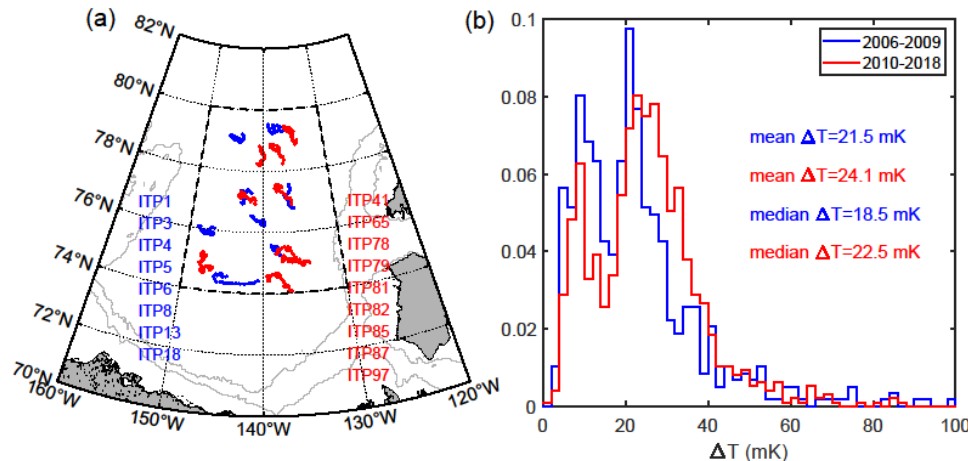


**Figure 9.** (a) ITP drift trajectories in the central Beaufort Gyre in May; (b) scaled histograms of the mixed layer temperature departure from
the local freezing point in the central BG in May.

Considering that the IMB observations do not catch the freezing and thawing of the seasonal sea ice, we also investigate

the decadal changes of the MO and FO using ULS data from three moorings in the BG (Moorings A,B, and D). Mooring C

is excluded due to its relatively short observation period. Since our earlier results indicate that the BMO in the BG occurs



approximately 17 days earlier than the SMO, thus the date, when the total ice thickness reached the annual maximum, obtained from the ULS data can be considered as the BMO. While, the date, when the total ice thickness reached the annual minimum, defined as the BFO may be debatable, due to simultaneous snow accumulation and basal melting. However, most of the IMB observations reveal that the snow depth was already relatively stable at the BFO around November–December. Thus, we consider it acceptable to consider the FO obtained from the ULS data as the BFO.

The observed ice thickness and calculated BMOs and BFOs during 2004–2018 are shown in Figures 10 and 11. For a comparison with results from the IMB observations presented above, all ULS BMOs and BFOs were divided into the same two periods of 2004–2009 and 2010–2018. The results revealed that the BMO advanced in all of the three moorings, but the advanced tendencies were insignificant at the 95% confidence level. The average BMO in mooring A was nearly the same in the two periods (23 May vs 22 May). At the same time, the average BMO in mooring B has been shifting to an earlier date from 10 June in 2004–2009 to 30 May in 2010–2018. The average BMO in mooring D also occurred earlier by 5 d from 1 June in 2007–2009 to 27 May in 2010–2018. The advance of the BMO obtained from the ULS data is consistent with the results obtained from the IMBs. However, the BFOs obtained by the two methods revealed a different change trend. The BFO in mooring A was delayed by 15 d from 25 September in 2004–2009 to 10 October in 2010–2017. In contrast, in the northernmost mooring B, which showed a later BFO in 2004 and 2006, the BFO advanced by 8 d from 8 October in 2004–2008 to 30 September in 2011–2017. In fact, since 2008, the BFO of mooring B also exhibited a delay. The BFO in mooring D was nearly the same between the two periods (29 September vs 30 September). Moorings A and D were located in the southern part of the BG, where the summer has been typically ice-free almost every year since 2007, except for 2013. When the sea ice melted completely, the BFO was almost the same as the SFO. Thus, the accelerated loss of sea ice and the frequent occurrence of ice-free summers in the BG may contribute to a later freeze-up.

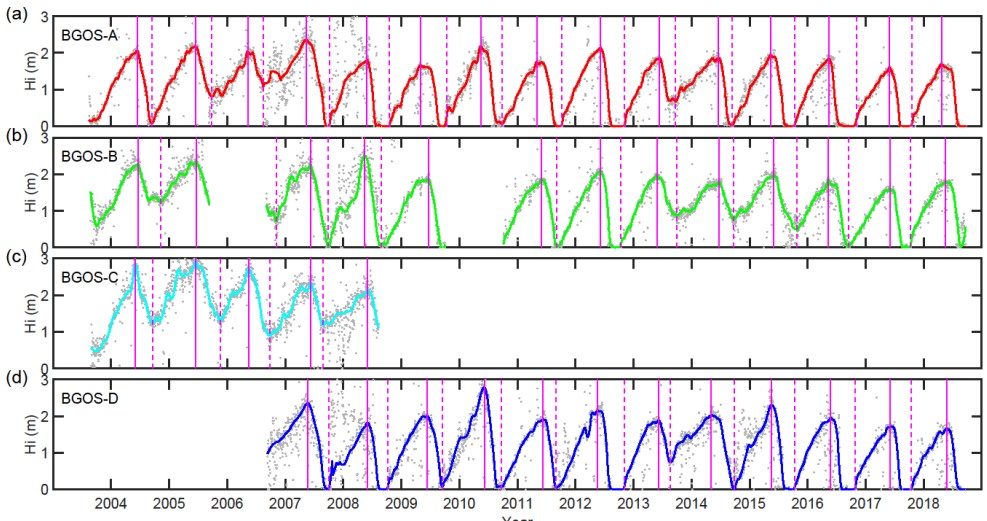

**Figure 10.** Time series of ice thickness observed by ULS in the BG (BGOS-A: red; BGOS-B: green; BGOS-C: cyan; BGOS-D: blue), gray dots denote the daily ice draft, with magenta solid and dash lines indicating the BMO and BFO, respectively.

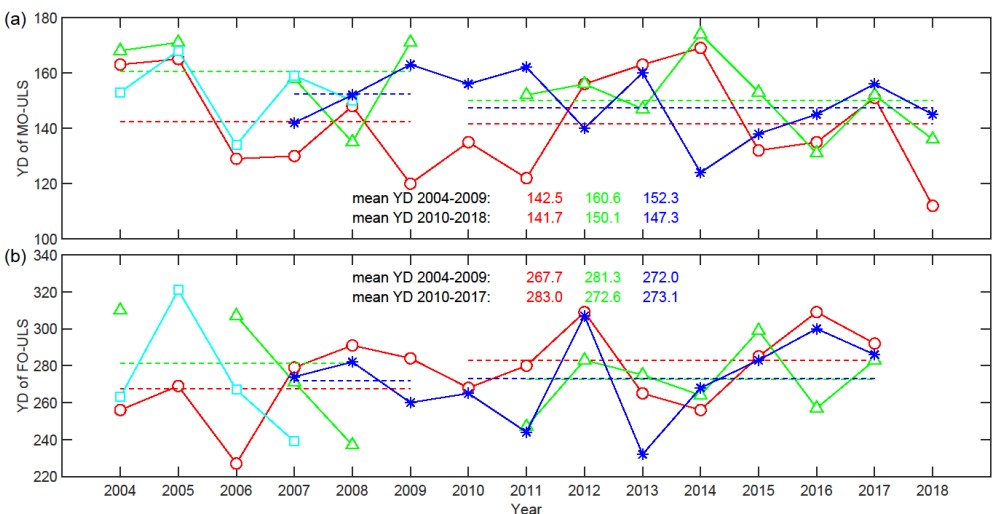

**Figure 11.** Interannual variations in (a) MO-ULS and (b) FO-ULS (BGOS-A: red; BGOS-B: green; BGOS-C: cyan; BGOS-D: blue), dashed lines denote the average of each period of 2004–2009 and 2010–2017.

**4 Conclusions**

In this study, we determined the timings of sea ice annual freeze-thaw cycles in the Beaufort Gyre and Central Arctic Ocean using multi-source data. Especially, the IMB observations provide the opportunity to investigate both the surface and basal freeze-thaw cycles of perennial sea ice, while ULS put emphasis on the freeze-thaw cycle of total sea ice thickness on seasonal

ice.

Based on multi-source observations, four pair of surface melt and freeze onsets are compared. The results reveal that both the remote sensing from PMW and SAT threshold method reliably capture the surface melt and freeze cycle quite efficiently when compared with surface mass balance observations from IMBs. The average BMOs were comparable in the CAO, and

approximately 17 days earlier than SMOs in the BG. While average BFOs were almost three months lagging behind SFOs for the entire Arctic Ocean.

During the transition of SMO, the topmost snow temperature increased to above melting point, indicating the initiation of surface melting. Reanalysis data indicated that the SMO was primarily driven by longwave radiation rather than shortwave radiation, but the mechanism for the SFO is the opposite. Synchronous underlying ocean observations revealed that the ice

bottom melt began when oceanic heat flux surpassed the upward conductive heat flux at sea ice bottom. While the ice basal freeze-up delay relative to the surface can be attribute to the regulation of heat capacity of sea ice itself, and the oceanic heat release from ocean mixed layer and subsurface layer. Ice cooling index determined by the near-surface air temperature, snow depth, and ice thickness shows a significant correlation with the temporal delay between BFO and SFO, with lower surface air temperature, thinner sea ice and snow cover tending to the earlier basal ice growth, and vice versa.

In the BG, the BMO derived from both Lagrangion IMB observations and Eulerian ULS observations exhibits an earlier trend, which can be ascribe to the earlier warming of surface ocean. In contrast, the BFO shows an earlier trend from Lagrangion IMB observations associated with the reduction of ice thickness of the multi-year ice with the buoy deployments, but a delay trend from Eulerian ULS observations because of the frequent occurrence of ice free summers for the south of BG region in recent years.

Note that, some limitation of our results should be considered. First, IMB only collected one-dimension point measurements of mass balance and are representative for the special ice floe where it was deployed. As a result, the melt and freeze onsets of other ice categories such as ponded ice, ridged ice, etc., are out of our scope. Second, the presences of ice interior melt, surface pond and false bottom, as well as the unfrozen cavities within the rubble of ridges greatly affected the energy budget, consequently the basal melt and freeze (Shestov et al., 2018; Provost et al., 2019, Smith et al., 2022). Third,

the majority of IMBs were deployed on multiyear undeformed ice (Planck et al., 2020), so the basal melt and freeze onsets of seasonal ice are under-represented. Compared to multiyear ice, seasonal ice has higher bulk brine, resulting in a smaller specific heat capacity and latent heat of fusion (Tucker et al., 1987; Wang et al., 2020), as well as a higher permeability during the summer (Lei et al., 2022), thus affecting the sea ice basal melt and freeze processes. Finally, due to the limited vertical observation range of ocean profile automatic observation instruments, some special processes near the ice bottom, such as
supercooling and false bottom were not characterized well.

Therefore, more intensive and elaborative ice mass balance observations of varieties ice types by IMB observations and other methods, and simultaneous upper ocean water properties observations in the future will vastly improve our capability to fully understand the ice-ocean system and the mass balance of sea ice in a changing Arctic.

## Data Availability

IMB data are publicly available at http://imb-crrel-dartmouth.org (last access on 4 Jan 2022). The ULS data are available from the Beaufort Gyre Exploration Program based at the Woods Hole Oceanographic Institution at http://www.whoi.edu/beaufortgyre (last access on 4 Jan 2022) in collaboration with researchers from Fisheries and Oceans Canada at the Institute of Ocean Sciences. The ITP data are available from the Ice-Tethered Profiler Program based at the Woods Hole Oceanographic Institution at https://www2.whoi.edu/site/itp (last access on 31 Dec 2021). Passive microwave

satellite data were downloaded from https://earth.gsfc.nasa.gov/cryo (last access on 31 Dec 2021). Sea ice concentration data were obtained from http://www.seaice.uni-bremen.de (last access on 31 Dec 2021). ERA5 reanalysis data was downloaded from the Research Data Archive of NCAR at https://rda.ucar.edu/ (last access on 3 Apr 2022).

## Author contributions

LL and RL undertook the data processing and analysis and prepared the manuscript. MH, DP, HH, and RL contributed to the
discussion during the writing process. All authors commented on the manuscript.

## Competing interests

The authors declare that they have no conflict of interest.

## Acknowledgments

We are very grateful to the IMB data bank, the Woods Hole Oceanographic Institution, the University of Bremen, and the
NASA Goddard Space Flight Center's Cryospheric Sciences Laboratory for providing the data of IMB, ITP, ice concentration, and ice freezing/melt onset.

## Financial support

This research has been supported by the National Key Research and Development Program (grant no. 2018YFA0605903),



the National Natural Science Foundation of China (grant nos. 42006037 and 41976219), the Natural Science Foundation of
Shanghai (grant no. 22ZR1468000), the US National Science Foundation NSF-2034919, and the US National Oceanic and
Atmospheric Administration NA200AR4310517.

**Review statement**

This paper was edited by XX and reviewed by XX anonymous referees.

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
