# Peer review of "Changes in the annual sea ice freeze-thaw cycle in the Arctic Ocean from 2001 to 2018"

_The Cryosphere, 2022_

## Referee Comment (RC3)

Review for tc-2022-137 "Changes in the annual sea ice freeze-thaw cycle in the Arctic Ocean from 2001 to 2018" writen by Lin et al..

The timing of melt onset , freeze onset and melt duration in Arctic plays a key role in the evolution of sea ice, and are important variables for understanding the Arctic climate system.  "Changes in the annual sea ice freeze-thaw cyle in the Arctic Ocean from 2001 to 2018 writen by Lin et al. studied the melt onset, freeze onset at the ice surface and at the ice bottom in the Beaufort Gyre and in the central Arcic Ocean from 2001-2018 using multi-source data, including sea ice autonous obeservations from mass balance buoys (IMBs), moored upward looking sonar (ULS), and Ice-Tethered Profilers (ITP), and remote sensing data from passive microwave  and reanalysis data from ECMWF ERA5. This manuscript is generally well writen, and the methodologies are proper for getting results in the mansucript. I suggest to publish the manusrcipt after minor revisions as listed below.

L32, add "," before "play a key role"

L37, L39, Please check if the references are cited properly.

L53, Add references for the statement in L50-53.

L54, "step" or "step-like"?

L64, can remove "due to freezing" after "basal ice growth".

L80, move  "freezing and thawing processes" before "sea ice surface"-

L81, It seems that "Another method" appears very sudden.  For this, L66-80 can be  listed as a seperate paragraph. In addition, please rewrite the sentences of L66-68 , for example, " The sea ice freeze-thaw cycle can be identified using the data measured by sea ice mass balance buoys (IMBs), which consist of a thermitor chain   in combination with … (Perovich et al. 2021). "

L110-114, rewrite the sentences to more concise, e.g., "The IMBs deployed on landast ice are excluded because shallow coastal waters have … (Eicken et al., 2005)".

L116, "The surface melt and freeze onsets is related to observations of near surface air temperature collected by the IMB" or "The surface melt and freeze onsets is related to the near surface air temperture", Please rewrite this sentence.

L144-146, If the 17 ITPs deployed in the central BG include the 12 ITPs used for detecting the basal melt or freeze?

L179, remove "of the" at the beginning of this line.

L223, Can you list the four pairs of surface melt and freeze onsets and one pair of basal melt and growth onsets in the list to make them more clear for the readers?

L226-228, It is better to clearly note 2014. Such as, "In 2014, ESMO-PMW was on 02 May, triggered by a spring storm event, and was about …SMO-IMB. Apart from that, the 2014 CSMO-PMW, SMO-SAT and SMO-IMB dovetail ….."

L233, add "2014 before "SMO-IMB", and before "ESMO-PMW".

L235, add "2015" before "SMO-IMB", and "2014" before "SFO-IMB".

L240, Figure 2. The legends for "ESFO-PMW, CSFO-PMW" in 2014 were wrong. Please correct them.

L255, remove "ensured" after "before fully freezing".

L257, For 13d delay after the SAT decreases below freezing, do you mean SFO-SAT?

L260, remove "defined as".L265, Figure 3 seems not clear. Please redraw it.

Explain "-" and "+" in the tables.

"median" should be "moderate", please correct it.

L273, please rechcek BFO lagged SFO by almost three or two months?

L276, confuse about "a decreasing trend of surface and basal melt onset" and " increasing trend of freeze onsets". Do you mean "surface and basal melt onsets later and freeze onsets earlier with an increase of latitute" or "surface and melt onsets are decreasing and the freeze onsets are increasing with the increase of latitute? Please rewrite.

L285, The meaning of "YD" in figure 4. For easy understanding, it is better to use date and month, e.g., 11 June, also in the figure, not Julian Day. Similar for Figure 8, and Figure 11.

L310, Please refer Table 2 in the text. The avearge surface net radiation changes were calculated using ERA5 reanalysis data. Please add this information to the Table 2.

L312-313, Difficult to understand this sentence, please rewrite it.

L314-315, "almost twice as large", compare to which?

L324-329, The trigger of SMO is an increase of longwave radiation (L300-301), and the key to BMO is when Fw becomes greater than Fc (L322). Thus the explanation in this paragraphy only can explain why the BMO in the BG occurred much earlier than the BMO in the CAO, but cannot explain why the BMO in the BG much eariler than the SMO, while they occurred almost at the same time in the CAO.

L346, change "," to ";" between two references.

L347-348, can be more clear or concise as " jointly resulted in the time of BFO later approximately 3 months than SFO.".

L361, "plays the opposite role of a thermal insulator" seems wrong. Please rewrite it.

L375, "Figure 5" should be "Figure 6".

L382, refer Table 4 in the text. The meaning of $\Delta H_i(m)$ in the Table?

L388, use different symbols for the surface and ice melt, to discrimate the symbols used in Eq. (5).

L396, remove "had "

L408, the meaning of (2.14m), (0.77m), (0.22m) ?

L410, what does the "intital" mean in "the initial ice thickness"? It means the ice thickness when the IMB installed, or the ice thickness when the SMO commences?

L412-413, Figure 6 indicate the linear relationship between ICI and BFO-SFO, which didn't consider the ice concentration. Thus this statement cannot related to Figure 6 as stated "relative to the linear regression as shown in Figure 6".

L421, add "Thus" before "The average annual ice thickness" to connect this sentence with the previous one.

L432, the caption of (d) and the figure ylabel is not same. Please correct the wrong one.

L435, Shoud "Figure 3c and 3d" be "Figure 4c and 4d"? recheck.

L443-444, The positive feedback between thinner and more vulneralbe ice and eariler BMO is a general statement. How can you derive this feedback from L435-443 where you talked about the decal changes of BMO and the average oceanic mixed layer temperture depature from the local freezing point.

L453: I don't agree "the IMB observations do not catch the freezing and thawing of  the seasonal sea ice".  This is also paradoxical your method for detecting SMO, SFO, BMO and BFO from IMB observations.

L468, "by  the two methods" or "by the  two moorings"? From the text, it seems that you mean "by the two moorings".

L480: Can remove BGOS-C: cyan since have excluded this mooring.

L498-499, Difficult to understand this sentence, please rewrite it.

---

## Author Comment (AC1)

Response to RC1

We authors thank you for your time and constructive comments on the manuscript "Changes in the annual sea ice freeze-thaw cycle in the Arctic Ocean from 2001 to 2018". We will consider each comment carefully and incorporate practically all of them.

General comments:

I have no general comments or concerns. The only two things I ask the authors to pay a bit more attention to is A) to relate their results more closely to the published literature to avoid the impression that the findings presented here are new throughout (the used set of observational data is but the results confirm published knowledge), and to B) reduce the usage of acronyms to a necessary minimum. I can understand the usage of SMO, SFO, BFO, BMO and the suffixes denoting the method / data used. But apart from that I find that a number of other acronyms might not be needed in the running text and would enhance readability of the paper a lot.

Reply: According to your comments, we will distinguish more clearer between the published knowledge and the new finding of our study, and refer to appropriate references for the publish literature in the manuscript, such as the thermal insulation of snow, the critical influence of the incoming shortwave radiation on sea ice melting. We will also like to reduce the usage of acronyms to a necessary minimum as your suggestion to make the paper more readable.

Specific comments:

L21: How does "sea ice cooling release heat"?
Reply: The ice temperature is heated by solar radiation in summer time. And the heat released from sea ice cooling can be divided into two parts. First is the sensible heat due to ice temperature change trigged by cold air temperature, which is described in the section 3.4. Second is the latent heat released due to the phase change from water to ice. The pore of ice could be filled with liquid melt water in summer season, which refreeze after the local ice temperature drops below freezing point. But this phenomenon is out of our scope because the limitation of IMB observation, which is briefly discussed in the section 4. We will rewrite the expression in abstract and results to make it more clearer.

L63-65: It is understandable that you are referring to space-borne altimetry here (simply because you are writing about basal ice melt and ice growth) and hence changing in ice thickness. But this is fundamentally different from how melt and freeze-onset is determined at the surface. I recommend that you i) make a comment about this fundamental difference and ii) provide the reasoning why it has to be this way.
Reply: Meltwater ponds accumulating on Arctic sea ice between May and September makes it difficult to differentiate between sea-ice and open-water leads, thus

prevented researchers from generating valid sea ice thickness observation in the summer months from any satellite sensor (Kwok et al., Elementa 2018). As a result, conventional algorithms have only enable sea ice thickness to be derived for the winter months of October to April using satellite remote sensing (Laxon et al., GRL 2013). We will point out this reason clearly to explain why the space-borne altimetry failed in the detection of melt and freeze onset.

L114/115: I am a bit concerned about the statement of an accuracy of 1 cm. This can certainly only be achieved if the sea ice floe in the area where the IMB is installed is nicely flat and has no deformation - particularly not at the ice bottom. In addition, while the surface is well defined - either as the bare ice surface or the snow surface, the ice underside can be rather blurry during freeze-up with congelation growth, can't it? I am therefore wondering whether the 1 cm given is a value reported from the lab or a value reported from field measurements.

Reply: According Planck et al. (2020), all the IMB were carefully chosen to deployed on the underformed level ice, and the resolution of acoustic sonars equipped on IMB was $\pm 1$ cm reported from the lab. Observations of ice bottom position from IMB did show some small fluctuations because the ice underside can the rather blurry within a certain area. However, the thermodynamic sea ice process is a slow-moving change. So, we used a 14-days moving filter for the detection of basal melt onset and freeze onset based on ice mass balance observation. To avoid misunderstanding, we will modify the sentence to make it more accurate by using "resolution of $\pm 1$ cm" instead of "accuracy of 1 cm".

L177: Is it correct that for SMO-IMB the SAT measured by the buoys are not filtered but used as they are - in contrast to method 2?

Reply: For SMO-IMB detection, we take surface mass balance observation as the dominant index, and surface air temperature as supplementary. Both thermodynamic (snow melting and snow accumulation) and dynamic (snow redistribution by wind forcing) processes can cause snow surface elevation change. In this paper, we only consider the thermodynamic of sea ice. So, the surface air temperature plays a subsidiary role to exclude the situation when surface elevation caused by dynamic processes. Some of the temperature fluctuations occurred in a short period might be erased by the 14-days moving filter. Thus, we used the SAT as they are when detecting the SMO-IMB.

L217: I am a bit concerned by neglecting the geostrophic current velocity. A value of 5 cm/s translates into 4.3 km / day which then is in the range of typical ice drift velocities. Also, since you use the difference of the two velocities in Equation (2), I don't quite get the motivation to neglect these cases. Wouldn't V be particularly large in case of a low geostrophic current velocity compared to an applicable ice drift velocity?

It might be helpful to further equation (2) and actually provide the equation with which you compute the friction speed which is then used in Equation (3). I imagine

that the issue of when you neglect which velocities becomes understandable better in that case.
Reply: We will perform the error analysis for neglecting the geostrophic current velocity when calculation the friction velocity to make the results more scientific. We will add ±5 cm/s to a typical ice velocity to calculate the range of friction velocity, as well as evaluate the impact on oceanic heat flux.

Since used numerical approximation method to compute the friction speed, there is no further equation than equation (2).

L256/257: Why is CSFO-PMW later than all the other products? What could be the reason?
Reply: We will compare the method of four pairs of surface freeze onset more carefully to figure out the reason why CSFO-PMW is later than all other products. We speculate that the observations of ice mass balance and surface air temperature only concentrate on a single point of an ice floe, while the observation of PMW represent a mean value within a certain area, which could also be influenced by melt ponds, leads, and open water.

L269-274: Would it make sense to also (or instead) provide the median quantities in order to minimize the influence of potential outliers?
It might make sense to rephrase the last sentence of this paragraph a bit such that it reads at the end: "... longer than at the surface, and was dominate ..."
Reply: Good point. We will check the median quantities of all onsets, and compare with the mean values to minimize the influence of potential outliers. The last sentence of this paragraph would also rephrase as your suggestion.

L280/281: "the ice" --> "sea ice" without "the". I am a bit surprized to see that the mean(?) sea ice thickness is larger in the BG than in the CAO.
Reply: The grammatical mistake will be corrected. The sea ice thickness does not strictly depend on latitudes. Actually, the thickest sea ice in Arctic Ocean is in the north of Canadian Arctic Archipelagos and Greenland. Most of the IMB deployed in the CAO was in the transpolar region. Beside, most multiyear ice represent by IMB was in the northern part of BG, which transferred from the north of Canadian Arctic Archipelagos. As a result, it is no surprise that sea ice thickness from IMB observation in BG was a little larger than it in the CAO.

L282/283: While there will certainly be mechanism that could have driven the observed scatter one should perhaps not forget that you are looking at data from quite a number of years with a certain variation in atmospheric and oceanographic conditions.
Reply: Yes, it is. We will point out this reality here. And we also discuss the mechanism in the section 3.4 and 3.5, where both atmospheric and oceanographic conditions are considered.

L298-300: Would it make sense to add to these net longwave radiation values before and after SMO the respective outgoing longwave radiation values computed following the Stefan-Boltzmann law, i.e. about 308 W/m^2 before SMO and about 316 W/m^2 after SMO, hence an increase in about 8 W/m^2? I believe it would make sense to dive a bit deeper into this and come up with an estimate of the actual increase in downwelling longwave radiation which - given the numbers we have at hand - seems to be from about 270 W/m^before SMO to close to 290 W/m^2 after SMO. This would fit much better to the statement made in the following sentence citing the work of Maksimovich and Vihma.

Reply: We will calculate the outgoing longwave radiation values with surface temperature following the Stefan-Boltzmann law. Since the surface air temperature is increasing, we could expect the increasing of outgoing longwave radiation, and much more increasing of downwelling longwave radiation compare to the increment of net longwave radiation. The result will fit much better to the statement made in the following sentence citing the work of Maksimovich and Vihma.

L376-381: Since these findings about the effect of snow insulation on sea-ice thickness in fall are not new I am suggesting that you back up these statements by a few references from the published literature to make clear that your results are in line with what has been published by other people.

Reply: We will refer to some published literature of the thermal insulation of snow layer on sea ice thickness.

L389 / Equation (6): I am puzzled about the usage of H_i and H_s first as sea ice thickness and snow thickness in the previous subsection while these are now used for "surface snow melt" and "surface ice melt" ... this reads a bit strange. Could it be that you want to refer to delta H_s and delta H_i, i.e. the amount by which the snow thickness and the sea ice thickness is reduced due to surface melt? In any case it would be good to use a different acronym or symbol to avoid confusion.

Reply: We will use "$\Delta H_i$ and $\Delta H_s$" instead of "$H_i$ and $H_s$" as your suggestion.

L409/411: "This suggests that the ... in summer" --> Also this finding is not new but simply confirms knowledge and results that has been published elsewhere and that should be referred to here.

Reply: We will add the reference of Stanton et al. (2012) to support the results that the basal sea ice melt is more likely related to the amount of solar heat input into the upper ocean in summer.

Stanton, T.P., Shaw, W. J., and Hutchings, J. K.: Observational study of relationships between incoming reaiation, open water fraction, and ocean-to-ice heat flux in the Transpolar Drift: 2002-2010. J. Geophys. Res., 117, C07005, doi:10.1029/2011JC007871, 2012.

L427/428: "We infer ..." I suggest to again add the aspect that this applies to the set of floes that were equipped with IMBs / ITPs. These floes were all at least second-year

ice floes (or at least becoming second year ice soon) and hence reflect - basically conditions of multiyear ice. Hence the statement made here might need to be limited to multiyear ice but does not apply to seasonal ice.

Reply: Yes, it is important. We will make the statement limited to multiyear ice.

Figure 10: In section 2.1.2 you give a description of the ULS data which, however, does not explain how you end up with the ice thickness data shown in this figure. Are these data also daily or are these filtered? I guess it would be good to share some more details here because it looks a bit weird the see ice draft values that are substantially larger than the sea ice thickness, for instance for BGOS-D in winter 2007/08.

"dash" --> "dashed" in the 2nd line of the caption.

Reply: It is a mistake in the caption. The "ice draft" in the caption is actually the "ice thickness". And the ice thickness data is converted from ULS ice draft data by scaling a reference density ration between sea water and sea ice, which will be stated in the data introduction. The grammatical mistake in caption will also be corrected.

Editoral remarks / Typos:

L42: I am wondering whether "delaying the ice recovery in winter" wouldn't fit better here than "suppressing the ice recovery ..."

Reply: We consider the "suppressing the ice recovery..."would fit better than "delaying the ice recovery in winter". The reason is "delaying the ice recovery in winter" only contains time delay, while "suppressing the ice recovery ..." also include the recovery of ice thickness.

L151: What kind of a grid is used here? EASE or polar-stereographic?

L162: What kind of a grid is used here?

Reply: Both the PMW data and sea ice concentration data are EASE-grid. We will make a clear statement in the data introduction.

L150/151: You could, similar to the ERA5 data, provide an URL and also the access data here.

L160: It is not really clear whether you applied the ASI algorithm yourself or whether you used product ready to download from somewhere. I suggest to clarify this issue and in case you downloaded the data from somewhere, again provide URL and access date.

Reply: All the URL are provided in the section of "Data Availability" in the end of manuscript. We will also introduce the sea ice concentration data a little deeper, including the algorithm.

L191-193: "The IMB observations ... Smith, et al., 2022)" --> Please check this sentence; I have difficulties to understand what you state here.

Reply: The sentence will be rewritten to make the expression more clearer. So, it is rewritten as "When false ice bottom exists, the IMB observation typically showed a sufficiently basal growth and following by a rapidly thinning in early to mid-summer without any significant atmospheric and oceanic temperature signals (Smith, et al, 2022).".

Figure 3: I recommend to enlarge this figure for better visibility of the written text.
Reply: Amplified the figure as suggestion.

It might make sense to indicate in the caption behind "day of the year" that you use the acronym "YD" in the panels themselves.
Reply: Redraw the Figure 4, and all the "YD" will be instead by month/day according to another reviewer's comments.

Figure 6: Please add information into the caption what is shown in the inset and therein also explain what the red asterisk is denoting.
Reply: Inset figure in figure 6 shows the roughly position during the time period between SFO and BFO, red asterisk denotes the north pole.

Figure 9 caption: What do you mean by "scaled"? What is the binsize used in panel b)?
Reply: "Scaled" refer to Toole et al. (2010). The binsize is 2 mK, which has been added in the caption of Figure 9.

Toole, J. M., Timmermans, M. L., Perovich, D. K., Krishfield, R. A., Proshutinsky, A., and Richter-Menge, J. A.: Influences of the ocean surface mixed layer and thermohaline stratification on Arctic Sea ice in the central Canada Basin, J. Geophys. Res. Oceans, 115(C10), doi:10.1029/2009JC005660, 2010.

L461-475: In this paragraph I recommend to early on note which of the moorings is located where because the respective map showing their locations is close to the beginning of the paper.
In addition, it might make sense to also mention which of the moorings is covered more likely by multiyear ice for at least some time of the year. I'd say is it C and D, followed by B and then A. Such a notion could also well back up your results.
Reply: We give the location of three moorings here again, and the first sentence of this paragraph is rewritten as "The observed ice thickness from three Moorings (A: 150 °W, 75 °N; B: 150 °W, 78 °N; D: 140 °W, 74 °N) and calculated BMOs and BFOs during 2004–2018 are shown in Figures 10 and 11.". The ice regions of these four mooring will also be introduced.

L18: "inconsistency" --> I am not convinced that this should be termed like this. I suggest to use "difference" and then try to find a replacement for the second usage of "difference" in the same sentence.
L20: "3" --> "three"

L23/L24: What is an "earlier trend"? I guess what you want to express that you observed "a trends towards earlier melt onset" and then "earlier trend" is not an adequate expression.

L25: "delayed trend" --> same comment as for "earlier trend" except that the direction in time is reversed here. What you want to state is that you found "a trend towards delayed onset of basal ice growth"

L98/99: "we evaluate the surface radiation" --> Does that mean that you evaluate (aka check the quality / validate) the reanalysis surface radiation data? If this is not the case, which I assume at the current state of the manuscript, then you might want to correct your formulation.

L142: Please check whether the acronym for the unit decibar is indeed "db" and not "dbar".

L204: "bulk conductive heat flux" --> perhaps add "in the sea ice"?

L276: Please write "negative trend" and "positive trend". A "decreasing trend" would be a trend value that changes over the associated unit, i.e. is first 10 days / latitude, becoming 5 days / latitude, for instance.

In general, it might make sense to instead of writing: "decreasing [negative] trend of surface and basal melt onset" something like "surface and basal melt onset dates becoming earlier" or "surface and basal melt onset shifting to earlier dates". Same suggestion applies to the "increasing [positive] trend.

L398: Is it okay to express a squared quantity with a negative sign? Wouldn't it perhaps be better to write "... a close negative correlation with ... $R^2 = 0.52$ ..."?

Table 1: Since "SMO-IMB" also utilizes SAT data you might want to add this information in the table.

Figure 4: I suggest to add information to the left 4 panels that allows one to see at a glimpse what the surface and what the basal data are. You could do this by vertically separating the two upper (a and b) from the two bottom (c and d) panels and write a title like "surface melt and freeze-up" just above panels a) and b) and "bottom melt and freeze-up" just above panels c) and d).

L369: "thinner by as much as < 0.50 m" --> Could you sharpen this statement a bit, please? > 0.50 m can mean everything from 0.51 m to 2 m or even more. Perhaps taking 0.5 as an approximate maximum value by which sea ice may thin between SFO and BFO would serve the purpose?

L418: "thinner ice" --> I guess you refer to YOUR cases of thinner ice, don't you. Please make this more clear.

L439: "an earlier trend of the BMO" --> perhaps better: a trend towards an earlier BMO"

L444-446: "However ..." --> I don't see the need for an "however" here. The fact that you observe an earlier BFO during the second half of your observation period with thinner sea ice (1.3 m) than during the first half with thicker sea ice (1.8 m) fits nicely into the picture. You could stress this by using "consequently" or "is in line with" ...

Conclusions: For the sake of readability I suggest to reduce usage of acronyms here to a necessary minimum and, for instance, always use the full name for the geographic locations.

L495-497: "While ... layer" --> I suggest to not begin this sentence with a "While". This is confusing.

In L496: "attribute" needs to be "attributed"

L500-504: See my notion on using the expression "earlier trend" made before.

L507: "presences" --> "presence"

L509: " ... 2022)" --> perhaps add: "but the effect these different conditions could have was not considered in our study."

Reply: All the grammatical mistakes and inappropriate expressions will be revised as your suggestion.

---

## Author Comment (AC2)

Response to RC2

We authors thank you for your time and constructive comments on the manuscript "Changes in the annual sea ice freeze-thaw cycle in the Arctic Ocean from 2001 to 2018". We would consider each comment carefully and incorporate practically all of them.

**Specific comments:**

L156-157: I would rephrase the definition of the continuous melt/freeze onset to the "the day after which ice surface melting/freezing conditions persist".

Reply: We will modify the definition of continuous melt/freeze onset as the suggestion.

L224: There is no Table S1.

Reply: Table S1 is in the supplement, which refer to the summary of surface and basal melt/freeze onsets from different methods.

L241: Paragraph 3.1 (Comparison of ice surface melt and freeze onsets from different methods): The period of time covered by this analysis is unclear. You mention that you use 55 IMB trajectories but not the time period.

Reply: Thanks for your comment. We check the time period of 55 IMB trajectories analyzed here, and the time period will be specified to "2002 to 2018" in the manuscript.

L265: I would replace Figure 3 by a table since half of the entries are empty.

Reply: We consider to retain the figure, since the colormap also shows the level of difference between 4 pairs of surface melt and freeze onsets.

L291: I would suggest using days instead of "d" here and throughout the manuscript

Reply: We will unify the units of "days" and "d" as "days" throughout the manuscript as your suggestion.

L308-309: Please rephrase "FO is primarily controlled by the decline of net shortwave radiation as the approaching of polar night". The sentence is confusing, not clear.

Reply: The point we want to express is that when the polar night approaching, the both the net shortwave radiation and net longwave radiation decreases. And the heat balance during before and after FO shows that the decline net shortwave radiation is the primarily controlling factor of FO. We will rewrite the sentence to make the expression clear.

L 310: Table 2 is not referenced in the text.
L381: Table 4 is not referenced in the text.
L334: you mention Figure 5b but there is no previous mention of Figure 5a.

Reply: we will add the reference of the Table 2 Table 4 and figure 5a in the text.

L314: Can you be a little more specific than "~several mK"?
Reply: we will modify the sentence refer to the original of Shaw et al. (2009), and specify "~several mK" to "within a few mK of the freezing point".

L331-332: I would rephrase "Here we further investigate the mechanism relevant to the BFO from the perspectives of both sea ice itself and underlying Ocean". It is unclear.
Reply: To make the expression more clearer, the sentence will be rewritten as "We further investigate the mechanism relevant to the time lag between BFO and SFO from the perspectives of both sea ice itself and underlying Ocean.".

L435: I believe you are referring to Figure 4 instead of Figure 3?
Reply: Thank you for your information. It is a mistake and. It should be "Figure 4c and 4d" instead of "Figure 3c and 3d".

L12: Replace "from surface" by "of the surface".
L94: Replace "show" by "showed".
L327: "absorbs" instead of "absorb".
L356: Delete "here".
L501: Replace "ascribe" by "attributed".
L502: replace "Lagrangion" by "Lagrangian".
L507-507: I would rephrase" Second, the presences of ice interior melt" to "Second, interior ice melt".
L516: Replace "varieties" by "diverse".
Reply: All the grammatical mistakes and inappropriate expressions will be revised as your suggestion.

---

## Author Comment (AC3)

We authors thank you for your time and constructive comments on the manuscript "Changes in the annual sea ice freeze-thaw cycle in the Arctic Ocean from 2001 to 2018". We would consider each comment carefully and incorporate practically all of them.

L37, L39, Please check if the references are cited properly.
Reply: We check the references, and consider them as proper citations.

L53, Add references for the statement in L50-53.
Reply: we will add the reference for the statement with (Persson, 2012).

L54, "step" or "step-like"?
Reply: We will change the "step" to "step-like".

L81, It seems that "Another method" appears very sudden. For this, L66-80 can be listed as a separate paragraph. In addition, please rewrite the sentences of L66-68, for example, "The sea ice freeze-thaw cycle can be identified using the data measured by sea ice mass balance buoys (IMBs), which consist of a thermitor chain in combination with … (Perovich et al. 2021)."
Reply: Thank you for your comments, which makes the context more logical. We will revise as your suggestion.

L110-114, rewrite the sentences to more concise, e.g., "The IMBs deployed on landfast ice are excluded because shallow coastal waters have … (Eicken et al., 2005)".
Reply: We will revise as your suggestion.

L116, "The surface melt and freeze onsets is related to observations of near surface air temperature collected by the IMB" or "The surface melt and freeze onsets is related to the near surface air temperature", Please rewrite this sentence.
Reply: Agree. "collect by the IMB" need to be deleted.

L144-146, If the 17 ITPs deployed in the central BG include the 12 ITPs used for detecting the basal melt or freeze?
Reply: The 17 ITPs deployed in the central BG was listed in the figure 9a, and the 12 ITPs used for detecting the basal melt or freeze were listed in Table 3. 7 of 12 ITPs used for detecting the basal melt or freeze were in BG, which were included in the 17 ITPs. And the rest of 5 ITP in CAO were not.

L223, Can you list the four pairs of surface melt and freeze onsets and one pair of basal melt and growth onsets in the list to make them more clear for the readers?
Reply: The four pairs of surface melt and freeze onsets and one pair of basal melt and

growth onsets have been listed in the supplement, due to their large space.

L257, For 13d delay after the SAT decreases below freezing, do you mean SFO-SAT?
Reply: Yes. the time when SAT decreases below freezing equals to SFO-SAT. The sentence will be rewritten as "… , with an average delay of 13 days after SFO-SAT."

L260, remove "defined as". L265, Figure 3 seems not clear. Please redraw it. Explain "-" and "+" in the tables.
"median" should be "moderate", please correct it.
Reply: We will delete "defined as" and revised from "median" to "moderate". The "-" and "+" in tables is the time difference of column minus row, which will be state in the caption of table.

L273, please rechcek BFO lagged SFO by almost three or two months?
Reply: The average SFO was on 20 August, and the average BFO was on 14 November. The BFO was 86 days lagged behind SFO, which is almost three months.

L276, confuse about "a decreasing trend of surface and basal melt onset" and "increasing trend of freeze onsets". Do you mean "surface and basal melt onsets later and freeze onsets earlier with an increase of latitude" or "surface and melt onsets are decreasing and the freeze onsets are increasing with the increase of latitude? Please rewrite.
Reply: To make the expression more clearer, the sentence will be rewritten as "The overall spatial patterns revealed that the surface and basal melt onsets were decreasing and the freeze onsets were increasing with the increase of latitude, as one would expect.".

L285, The meaning of "YD" in figure 4. For easy understanding, it is better to use date and month, e.g., 11 June, also in the figure, not Julian Day. Similar for Figure 8, and Figure 11.
Reply: For easy understanding, we will modify the time in figure 4, figure 8, and figure 11, i.ie., using "MM/DD" instead of "YD", as well as throughout of the manuscript.

L310, Please refer Table 2 in the text. The average surface net radiation changes were calculated using ERA5 reanalysis data. Please add this information to the Table 2.
Reply: The reference of Table 2 will be added, as well as the information in the title.

L312-313, Difficult to understand this sentence, please rewrite it.
Reply: we compare the average $\Delta T$, $u_{*0}$, $F_w$, and $F_c$ between the time period of 10 days before BMO and 10 days after BMO using data of 12 ITPs, this sentence will be rewritten to make it easy for understanding.

L314-315, "almost twice as large", compare to which?

Reply: We will rewrite the sentence as "… almost twice as large as $\Delta T$ (–10 d),…"

L324-329, The trigger of SMO is an increase of longwave radiation (L300-301), and the key to BMO is when Fw becomes greater than Fc (L322). Thus the explanation in this paragraph only can explain why the BMO in the BG occurred much earlier than the BMO in the CAO, but cannot explain why the BMO in the BG much earlier than the SMO, while they occurred almost at the same time in the CAO.
Reply: The SMO and BMO depend on the heat budget of ice surface and base, respectively. In addition to the larger amount of incoming solar radiation based on latitudes, the larger fraction of leads also results in more absorption of solar radiation. The last sentence in this paragraph used "may" and "partly" as a speculation.

L347-348, can be more clear or concise as "jointly resulted in the time of BFO later approximately 3 months than SFO.".
Reply: it will be revised as your suggestion, which make the expression more clear.

L361, "plays the opposite role of a thermal insulator" seems wrong. Please rewrite it.
Reply: The snow cover insulate the conductive heat transfer due to its lower thermal diffusivity. So thicker snow depth plays the opposite role when compared with lower surface air temperature. we will rewrite the sentence as "… plays the opposite role as a thermal insulator…".

L382, refer Table 4 in the text. The meaning of ΔHi(m) in the Table?
Reply: Table 4 will be referred in the text and the column of ΔHi(m) could be deleted.

L388, use different symbols for the surface and ice melt, to discrimate the symbols used in Eq. (5).
Reply: We will change $H_i$ and $H_s$ to $\Delta H_i$ and $\Delta H_s$ according to another reviewer's comment.

L408, the meaning of (2.14m), (0.77m), (0.22m) ?
Reply: These numbers show the exact total basal melt.

L410, what does the "initital" mean in "the initial ice thickness"? It means the ice thickness when the IMB installed, or the ice thickness when the SMO commences?
Reply: Here, the "initial" in "initial ice thickness" means the time when surface melt begin. So, the initial ice thickness is the ice thickness when the SMO commences. We will specify this in the text.

L412-413, Figure 6 indicate the linear relationship between ICI and BFO-SFO, which didn't consider the ice concentration. Thus this statement cannot related to Figure 6 as stated "relative to the linear regression as shown in Figure 6".
Reply: We will delete this inappropriate sentence.

L443-444, The positive feedback between thinner and more vulnerable ice and earlier BMO is a general statement. How can you derive this feedback from L435- 443 where you talked about the decal changes of BMO and the average oceanic mixed layer temperature departure from the local freezing point.

Reply: The changes of temperature departure from the local freezing point in May derived from ITP observation reveals that the surface ocean is warmed earlier than the past, which mainly attributed to solar absorption through opening leads. Thus, there is a positive feedback between thinner and more vulnerable ice and earlier BMO. We will add the impact of opening leads on the changes of oceanic mixed layer temperature.

L453: I don't agree "the IMB observations do not catch the freezing and thawing of the seasonal sea ice". This is also paradoxical your method for detecting SMO, SFO, BMO and BFO from IMB observations.

Reply: The key point of this sentence is the "seasonal sea ice", since the majority of IMB were deployed on the multiyear ice. That is the reason why we also used the ULS data to represent the seasonal sea ice.

L468, "by the two methods" or "by the two moorings"? From the text, it seems that you mean "by the two moorings".

Reply: The two methods mean two data sources, i.e., IMB and ULS. So we will revise to "…from IMB and ULS…".

L498-499, Difficult to understand this sentence, please rewrite it.

Reply: It should be the "earlier basal growth onset" instead of "earlier basal growth". We will rewrite the sentence.

L32, add "," before "play a key role"
L64, can remove "due to freezing" after "basal ice growth".
L80, move "freezing and thawing processes" before "sea ice surface"-
L179, remove "of the" at the beginning of this line.
L226-228, It is better to clearly note 2014. Such as, "In 2014, ESMO-PMW was on 02 May, triggered by a spring storm event, and was about ...SMO-IMB. Apart from that, the 2014 CSMO-PMW, SMO-SAT and SMO-IMB dovetail ….."
L233, add "2014 before "SMO-IMB", and before "ESMO-PMW".
L235, add "2015" before "SMO-IMB", and "2014" before "SFO-IMB".
L240, Figure 2. The legends for "ESFO-PMW, CSFO-PMW" in 2014 were wrong. Please correct them.
L255, remove "ensured" after "before fully freezing".
L346, change "," to ";" between two references.
L375, "Figure 5" should be "Figure 6".
L396, remove "had"
L421, add "Thus" before "The average annual ice thickness" to connect this sentence

with the previous one.

L432, the caption of (d) and the figure ylabel is not same. Please correct the wrong one.

L435, Should "Figure 3c and 3d" be "Figure 4c and 4d"? recheck.

L480: Can remove BGOS-C: cyan since have excluded this mooring.

Reply: All the grammatical mistakes and inappropriate expressions will be revised as your suggestion.

---

## Author Response (AR1)

The authors thank the reviewers for their time and constructive comments on the manuscript "Changes in the annual sea ice freeze-thaw cycle in the Arctic Ocean from 2001 to 2018". We carefully considered each comment, and made corresponding changes based on the feedback provided in all cases.

**Response to RC1**

General comments:

I have no general comments or concerns. The only two things I ask the authors to pay a bit more attention to is A) to relate their results more closely to the published literature to avoid the impression that the findings presented here are new throughout (the used set of observational data is but the results confirm published knowledge), and to B) reduce the usage of acronyms to a necessary minimum. I can understand the usage of SMO, SFO, BFO, BMO and the suffixes denoting the method / data used. But apart from that I find that a number of other acronyms might not be needed in the running text and would enhance readability of the paper a lot.

Reply: According to your comments, we will distinguish more clear between the published knowledge and the new finding of our study, and refer to appropriate references for the publish literature in the manuscript. For example, the thermal insulation of snow was referred to Ledley (1991). the critical influence of the incoming shortwave radiation on sea ice melting was referred to Stanton et al. (2012).

We agree that the extensive use of acronyms might hamper readability. So we reduce the usage of acronyms to a necessary minimum as suggested to make the paper more readable. We changed the "BG" and "CAO" to "Beaufort Gyre" and "the Central Arctic Ocean", respectively, in conclusion. We also change the "MO" and "FO" to "melt onset" and "freeze onset", respectively, throughout the manuscript.

Ledley, T. S.: Snow on sea ice: Competing effects in shaping climate. J. Geophys. Res. Atmos, 96(D9), 17195-17208, https://doi.org/10.1029/91JD01439, 1991.
Stanton, T.P., Shaw, W. J., and Hutchings, J. K.: Observational study of relationships between incoming reaiation, open water fraction, and ocean-to-ice heat flux in the Transpolar Drift: 2002-2010. J. Geophys. Res., 117, C07005, doi:10.1029/2011JC007871, 2012.

Specific comments:

L21: How does "sea ice cooling release heat"?
Reply: The wording was poorly chosen. We changed this to "This temporal delay was caused by a combination of cooling the sea ice, the ocean mixed layer, and the ocean subsurface layer". (Line 21)

L63-65: It is understandable that you are referring to space-borne altimetry here (simply because you are writing about basal ice melt and ice growth) and hence changing in ice thickness. But this is fundamentally different from how melt and

freeze-onset is determined at the surface. I recommend that you i) make a comment about this fundamental difference and ii) provide the reasoning why it has to be this way.

Reply: Meltwater ponds accumulating on Arctic sea ice between May and September make it difficult to differentiate between sea-ice and open-water leads, thus preventing researchers from generating valid sea ice thickness observations in the summer months from any satellite sensor (Kwok et al., 2018). As a result, conventional algorithms have only enabled sea ice thickness to be derived for the winter months of October to April using satellite remote sensing (Laxon et al., 2013).

We point out that the complete freeze-thaw cycle can not be obtained with any satellite altimetry data because of the data gaps in the melt season (May to September). And we also briefly explained the reason, which is the difficulty to differentiate between sea ice and open-water leads with the impact of melt ponds. The sentence was rewritten as "Despite their importance, the complete freeze-thaw cycle, as well as the onset of basal ice melt and basal ice growth (BMO and BFO), cannot be directly determined by any remotely-sensed radar or laser altimeter because of the difficulty to differentiate between sea ice and open-water leads with the impact of melt ponds in the summer melt season (Laxon et al., 2013; Kwok et al., 2018).". (Line 64-67)

Laxon, S. W., Giles, K. A., Ridout, A. L., Wingham, D. J., Willatt, R., Cullen, R., Kwok, R., Schweiger, A., Zhang, J., Haas, C., Hendricks, S., Krishfield, R., Kurtz, N., Farrell S., and Davidson, M.: CryoSat-2 estimates of Arctic sea ice thickness and volume, Geophys. Res. Lett., 40, 732–737, doi:10.1002/grl.50193, 2013.

Kwok, R., Cunningham, G. F., and Armitage, T. W. K.: Relationship between specular returns in CryoSat-2 data, surface albedo, and Arctic summer minimum ice extent. Elem Sci Anth, 6: 53. DOI: https://doi.org/10.1525/elementa.311, 2018.

L114/115: I am a bit concerned about the statement of an accuracy of 1 cm. This can certainly only be achieved if the sea ice floe in the area where the IMB is installed is nicely flat and has no deformation - particularly not at the ice bottom. In addition, while the surface is well defined - either as the bare ice surface or the snow surface, the ice underside can be rather blurry during freeze-up with congelation growth, can't it? I am therefore wondering whether the 1 cm given is a value reported from the lab or a value reported from field measurements.

Reply: The accuracy of acoustic sonars equipped on IMB is ±5 mm (Richter-Menge et al., 2006), and the resolution is 1 cm (Planck et al., 2020). According to Planck et al. (2020), all IMBs were deployed on undeformed level ice. Although observations of ice bottom position from IMB did show some small fluctuations because the ice underside can the rather blurry within a certain area. The thermodynamic sea ice process is a slow-moving change. So, we used a 14-days moving filter for the detection of basal melt onset and freeze onset based on ice mass balance observation.

We rewrote the sentence to "The acoustic sounders on the IMBs measure the distance to the ice surface and ice base with a resolution of ±1 cm (Planck et al., 2020).". (Lines 116-117)

Richter-Menge, J. A., Perovich, D. K., Elder, B. C., Claffey, K., Rigor, I., and Ortmeyer, M.: Ice mass-balance buoys: a tool for measuring and attributing changes in the thickness of the Arctic sea-ice cover, Ann. Glaciol., 44, 205-210, 2006.

Planck, C. J., Perovich, D. K., and Light, B.: A synthesis of observations and models to assess the time series of sea ice mass balance in the Beaufort Sea, J. Geophys. Res. Oceans, 125, e2019JC015833, doi:10.1029/2019JC015833, 2020.

L177: Is it correct that for SMO-IMB the SAT measured by the buoys are not filtered but used as they are - in contrast to method 2?

Reply: For the SMO-IMB detection, we take the surface mass balance observation as the dominant index, and surface air temperature as a supplement. Both thermodynamic (snow melting and snow accumulation) and dynamic (snow redistribution by wind forcing) processes can cause a snow surface elevation change. In this paper, we only consider the thermodynamics of sea ice. So, the surface air temperature plays a subsidiary role to exclude the situation when a surface elevation change is caused by dynamic processes. Some of the short-term temperature fluctuations might be smoothed by the 14-days moving filter. Thus, we used the SAT as they are (unfiltered) when detecting the SMO-IMB.

L217: I am a bit concerned by neglecting the geostrophic current velocity. A value of 5 cm/s translates into 4.3 km / day which then is in the range of typical ice drift velocities. Also, since you use the difference of the two velocities in Equation (2), I don't quite get the motivation to neglect these cases. Wouldn't V be particularly large in case of a low geostrophic current velocity compared to an applicable ice drift velocity?
It might be helpful to further equation (2) and actually provide the equation with which you compute the friction speed which is then used in Equation (3). I imagine that the issue of when you neglect which velocities becomes understandable better in that case.

Reply: The calculation of friction velocity is according to McPhee et al. (2003). And "$V$" is the difference between ice velocity relative to surface geostrophic flow. On short time scales associated with individual storm events, ice drift velocity usually far exceeds the geostrophic ocean current, and $V$ is assumed as the actual ice velocity.

Here, we performed the error analysis for neglecting the geostrophic current velocity when calculating the friction velocity. The ice drift velocity on BMO range from 4 cm s$^{-1}$ to 17 cm s$^{-1}$, with a mean value of 8 cm s$^{-1}$. Thus, the maximum uncertainty of the heat flux when neglecting the geostrophic current will be 3.0 W m$^{-2}$ for a mean $\Delta T$ of 40 mK. Actually, the maximum surface geostrophic flow mainly

occurs in the marginal seas and Fram Strait, which is outside of our study region. The majority of the surface geostrophic flows in BG and CAO are far below 5 cm s$^{-1}$ (Timmermans and Marshall, 2020). In addition, the results of the BMO, on the other hand, proves than the heat budget of the ice-ocean interface is reasonable. Therefore, we believe that neglecting the geostrophic current velocity when calculating the friction velocity is acceptable.

We used a numerical approximation method to compute the friction speed. So, there is no further equation than equation (2).

Timmermans, M.-L., and Marshall, J.: Understanding Arctic Ocean circulation: A review of ocean dynamics in a changing climate. J. Geophys. Res.: Oceans, 125, e2018JC014378. https://doi.org/10.1029/2018JC014378, 2020.

L256/257: Why is CSFO-PMW later than all the other products? What could be the reason?
Reply: We compared the method of four pairs of surface freeze onset, and speculated that the reason might be the spatial resolution. The observations of ice mass balance and surface air temperature only represented a single point of the ice floe. In contrast, the observation of the PMW was a mean value within a certain footprint area, which could also be influenced by the liquid water from melt ponds, leads, and open water. We added this speculation to the corresponding text passage. (Line 261-264)

L269-274: Would it make sense to also (or instead) provide the median quantities in order to minimize the influence of potential outliers?
It might make sense to rephrase the last sentence of this paragraph a bit such that it reads at the end: "... longer than at the surface, and was dominate ..."
Reply: Good point. We checked the median values of all onsets, and compared this with the mean values to minimize the influence of potential outliers. We added the median day after the mean value in the text (Lines 277-280). The results showed that the difference between median value and mean value was no more than 2 days. So we used the mean value throughout our paper. The last sentence of this paragraph was rephrased according to the suggestion. (Line 282)

L280/281: "the ice" --> "sea ice" without "the".
Reply: Changes as suggested. (lines 289)

I am a bit surprised to see that the mean(?) sea ice thickness is larger in the BG than in the CAO.
Reply: The sea ice thickness does not strictly depend on latitudes. Actually, the thickest sea ice in the Arctic Ocean is found to the north of the Canadian Arctic Archipelago and Greenland. Most of the IMBs deployed in the CAO were in the transpolar drift region. Additionally, most multiyear ice was in the northern part of the BG, which was transported there from the north of the Canadian Arctic Archipelagos.

As a result, it is no surprise that sea ice thickness from IMB observation in the BG was a little larger than it in the CAO.

L282/283: While there will certainly be mechanism that could have driven the observed scatter one should perhaps not forget that you are looking at data from quite a number of years with a certain variation in atmospheric and oceanographic conditions.
Reply: Yes, this is certainly correct. We pointed out this reality here (Lines 283-284), and this is also discussed in more detail in Sections 3.4 and 3.5, where both atmospheric and oceanographic conditions are considered.

L298-300: Would it make sense to add to these net longwave radiation values before and after SMO the respective outgoing longwave radiation values computed following the Stefan-Boltzmann law, i.e. about 308 W/m^2 before SMO and about 316 W/m^2 after SMO, hence an increase in about 8 W/m^2? I believe it would make sense to dive a bit deeper into this and come up with an estimate of the actual increase in downwelling longwave radiation which - given the numbers we have at hand - seems to be from about 270 W/m^2 before SMO to close to 290 W/m^2 after SMO. This would fit much better to the statement made in the following sentence citing the work of Maksimovich and Vihma.
Reply: We calculated the outgoing longwave radiation values with surface temperature following the Stefan-Boltzmann law. The results show an increase of 7.8 W m$^{-2}$ when comparing between 10 days before and after the SMO. This corresponds to a total increase of 19.1 W m$^{-2}$ for the downward longwave radiation, which fits much better to the statement made in the following sentence citing the work of Maksimovich and Vihma (2012) and Persson (2012). We added the calculation of upward longwave radiation in the text. (Lines 308-310)

L376-381: Since these findings about the effect of snow insulation on sea-ice thickness in fall are not new I am suggesting that you back up these statements by a few references from the published literature to make clear that your results are in line with what has been published by other people.
Reply: we added a reference in Line 376, for the thermal insulation of snow.

Ledley, T. S.: Snow on sea ice: Competing effects in shaping climate. J. Geophys. Res. Atmos, 96(D9), 17195-17208, https://doi.org/10.1029/91JD01439, 1991.

L389 / Equation (6): I am puzzled about the usage of H_i and H_s first as sea ice thickness and snow thickness in the previous subsection while these are now used for "surface snow melt" and "surface ice melt" ... this reads a bit strange. Could it be that you want to refer to delta H_s and delta H_i, i.e. the amount by which the snow thickness and the sea ice thickness is reduced due to surface melt? In any case it would be good to use a different acronym or symbol to avoid confusion.

Reply: We now use "$\Delta H_i$ and $\Delta H_s$" instead of "$H_i$ and $H_s$" as suggested. (Lines 403-404)

L409/411: "This suggests that the ... in summer" --> Also this finding is not new but simply confirms knowledge and results that has been published elsewhere and that should be referred to here.
Reply: We added the reference of Stanton et al. (2012) to support the results that the basal sea ice melt is more likely related to the amount of solar heat input into the upper ocean in summer. (Line 426)

Stanton, T.P., Shaw, W. J., and Hutchings, J. K.: Observational study of relationships between incoming reaiation, open water fraction, and ocean-to-ice heat flux in the Transpolar Drift: 2002-2010. J. Geophys. Res., 117, C07005, doi:10.1029/2011JC007871, 2012.

L427/428: "We infer ..." I suggest to again add the aspect that this applies to the set of floes that were equipped with IMBs / ITPs. These floes were all at least second-year ice floes (or at least becoming second year ice soon) and hence reflect - basically conditions of multiyear ice. Hence the statement made here might need to be limited to multiyear ice but does not apply to seasonal ice.
Reply: Yes, good point. We added a statement referring to multiyear ice as suggested. (Line 441)

Figure 10: In section 2.1.2 you give a description of the ULS data which, however, does not explain how you end up with the ice thickness data shown in this figure. Are these data also daily or are these filtered? I guess it would be good to share some more details here because it looks a bit weird the see ice draft values that are substantially larger than the sea ice thickness, for instance for BGOS-D in winter 2007/08.
Reply: This was a mistake in the caption. The "ice draft" in the caption is actually the "ice thickness". And the ice thickness data is converted from ULS ice draft data by scaling a reference density ration between sea water and sea ice. We stated this in the introduction of the ULS data. (Lines 136-137)
"dash" --> "dashed" in the 2nd line of the caption.
Reply: Changes as suggested. (Lines 493)

Editorial remarks / Typos:

L18: "inconsistency" --> I am not convinced that this should be termed like this. I suggest to use "difference" and then try to find a replacement for the second usage of "difference" in the same sentence.
Reply: We revised "inconsistency" to "difference" as suggested, and used "distinct" instead of "difference in the" in the later part of this sentence. (Line 18)

L20: "3" --> "three"
Reply: Revised as suggested. (Line 20)

L23/L24: What is an "earlier trend"? I guess what you want to express that you observed "a trends towards earlier melt onset" and then "earlier trend" is not an adequate expression.
L25: "delayed trend" --> same comment as for "earlier trend" except that the direction in time is reversed here. What you want to state is that you found "a trend towards delayed onset of basal ice growth"
L439: "an earlier trend of the BMO" --> perhaps better: a trend towards an earlier BMO"
L500-504: See my notion on using the expression "earlier trend" made before.
Reply: We revised the expression as suggested throughout the manuscript. Lines (23-25; 49; 453; 518-519; 521)

L42: I am wondering whether "delaying the ice recovery in winter" wouldn't fit better here than "suppressing the ice recovery ..."
Reply: We consider the "suppressing the ice recovery..."would fit better than "delaying the ice recovery in winter". The reason is that "delaying the ice recovery in winter" only suggests a time delay, while "suppressing the ice recovery ..." also includes the recovery of ice thickness.

L98/99: "we evaluate the surface radiation" --> Does that mean that you evaluate (aka check the quality / validate) the reanalysis surface radiation data? If this is not the case, which I assume at the current state of the manuscript, then you might want to correct your formulation.
Reply: we revised "…evaluate…" to "…investigate the changes of…". (Line100)

L142: Please check whether the acronym for the unit decibar is indeed "db" and not "dbar".
Reply: we revised the "db" to "dbar". (Line 145)

L151: What kind of a grid is used here? EASE or polar-stereographic?
L162: What kind of a grid is used here?
Reply: Both the PMW data and sea ice concentration data are EASE-grid. We included a statement in the data introduction. (Line 154, 165)

L150/151: You could, similar to the ERA5 data, provide an URL and also the access data here.
Reply: Added. (Line 154)

L160: It is not really clear whether you applied the ASI algorithm yourself or whether you used product ready to download from somewhere. I suggest to clarify this issue and in case you downloaded the data from somewhere, again provide URL and access date.
Reply: We rewrote the first sentence of the sea ice concentration paragraph to indicate that it is downloaded from website, and added the URL and access date. (Lines 163-165)

L191-193: "The IMB observations ... Smith, et al., 2022)" --> Please check this sentence; I have difficulties to understand what you state here.
Reply: The sentence was rewritten to "When false ice bottoms exist, the IMB observations typically showed basal growth without any associated atmospheric and/or oceanic temperature signals, followed by a rapid thinning in early to mid-summer (Smith, et al, 2022).". (Lines 194-196)

Table 1: Since "SMO-IMB" also utilizes SAT data you might want to add this information in the table.
Reply: Added as suggested. (Table 1)

L204: "bulk conductive heat flux" --> perhaps add "in the sea ice"?
Reply: Added as suggested. (Lines 207-208)

Figure 3: I recommend to enlarge this figure for better visibility of the written text.
Reply: The figure was enlarged as suggested. We also indicated the meaning of "+" and "−" as per another reviewer's suggestion. (Figure 3; Line 273)

L276: Please write "negative trend" and "positive trend". A "decreasing trend" would be a trend value that changes over the associated unit, i.e. is first 10 days / latitude, becoming 5 days / latitude, for instance.
In general, it might make sense to instead of writing: "decreasing [negative] trend of surface and basal melt onset" something like "surface and basal melt onset dates becoming earlier" or "surface and basal melt onset shifting to earlier dates". Same suggestion applies to the "increasing [positive] trend.
Reply: Revised as suggested. (Lines 284-285)

Figure 4: I suggest to add information to the left 4 panels that allows one to see at a glimpse what the surface and what the basal data are. You could do this by vertically separating the two upper (a and b) from the two bottom (c and d) panels and write a title like "surface melt and freeze-up" just above panels a) and b) and "bottom melt and freeze-up" just above panels c) and d).
Reply: We added "SMO", "SFO", "BMO", "BFO" in the panels a, b, c, d, respectively. (Figure 4)
It might make sense to indicate in the caption behind "day of the year" that you use the acronym "YD" in the panels themselves.

Reply: We redid Figure 4, and all the "YD"s were revised as month/day according to another reviewer's comments. (Figure 4)

L369: "thinner by as much as < 0.50 m" --> Could you sharpen this statement a bit, please? > 0.50 m can mean everything from 0.51 m to 2 m or even more. Perhaps taking 0.5 as an approximate maximum value by which sea ice may thin between SFO and BFO would serve the purpose?
Reply: We revised this to "… as much as 0.63 m compared to that at the SFO (IMB 2011K).". (Line 383)

Figure 6: Please add information into the caption what is shown in the inset and therein also explain what the red asterisk is denoting.
Reply: The insert in Figure 6 shows the approximate position during the time period between SFO and BFO, and the red asterisk denotes the north pole. This information was added to the caption. (Figure 6)

L398: Is it okay to express a squared quantity with a negative sign? Wouldn't it perhaps be better to write "... a close negative correlation with ... $R^2 = 0.52$ ..."?
Reply: This was a mistake, and it should have been "R" instead of "$R^2$". We corrected the mistake throughout the manuscript (Line 389; 392; 407; 412; 428; 429), as well as in Figure 6 and Figure7.

L418: "thinner ice" --> I guess you refer to YOUR cases of thinner ice, don't you. Please make this more clear.
Reply: We rewrote the sentence as "As investigated above with IMB observations, basal growth of thinner sea ice started earlier compared to thicker ice.". (Line 430)

L444-446: "However ..." --> I don't see the need for an "however" here. The fact that you observe an earlier BFO during the second half of your observation period with thinner sea ice (1.3 m) than during the first half with thicker sea ice (1.8 m) fits nicely into the picture. You could stress this by using "consequently" or "is in line with" ...
Reply: The sentence was rewritten as "The average BFO was on 15 November in 2010–2018 (15 cases), which was 8 days earlier than (23 November) in 2001–2009 (9 cases), and which is in line with thinner ice favoring an earlier onset of basal ice growth (1.30 m vs 1.83 m).". (Lines 457-459)

Figure 9 caption: What do you mean by "scaled"? What is the bin size used in panel b)?
Reply: "Scaled" means the ratio after standardization referred to Toole et al. (2010). We added the y-axis label "PDF" in Figure 9 (b). The bin size is 2 mK, which has been added in the caption of Figure 9.

Toole, J. M., Timmermans, M. L., Perovich, D. K., Krishfield, R. A., Proshutinsky, A., and Richter-Menge, J. A.: Influences of the ocean surface mixed layer and

thermohaline stratification on Arctic Sea ice in the central Canada Basin, J. Geophys. Res. Oceans, 115(C10), doi:10.1029/2009JC005660, 2010.

L461-475: In this paragraph I recommend to early on note which of the moorings is located where because the respective map showing their locations is close to the beginning of the paper.
In addition, it might make sense to also mention which of the moorings is covered more likely by multiyear ice for at least some time of the year. I'd say is it C and D, followed by B and then A. Such a notion could also well back up your results.
Reply: We indicated the locations of the three moorings here again, and the first sentence of this paragraph is now rewritten as "The observed ice thickness from three Moorings (A: 150 ºW, 75 ºN; B: 150 ºW, 78 ºN; D: 140 ºW, 74 ºN) and calculated BMOs and BFOs during 2004–2018 are shown in Figures 10 and 11.". (Lines 475-476)
    Except 2013 for all three moorings and 2014 and 2015 for mooring B, the sea ice melted completely in the region after 2007.

Conclusions: For the sake of readability I suggest to reduce usage of acronyms here to a necessary minimum and, for instance, always use the full name for the geographic locations.
Reply: Revised the "BG" and "CAO" to "Beaufort Gyre" and "Central Arctic Ocean". (Conclusion)

L495-497: "While ... layer" --> I suggest to not begin this sentence with a "While". This is confusing.
Reply: Delete "While". (Line 513)

In L496: "attribute" needs to be "attributed"
Reply: Revised as suggested. (Line 513)

L507: "presences" --> "presence"
Reply: The sentence was rewritten as "Second, interior ice melt, …" according to another reviewer's suggestion. (Lines 525)

L509: " ... 2022)" --> perhaps add: "but the effect these different conditions could have was not considered in our study."
Reply: Added as suggested. (Lines 527-528)

**Response to RC2**

Specific comments:

L12: Replace "from surface" by "of the surface".

Reply: Revised as suggested. (Line 12)

L94: Replace "show" by "showed".
Reply: Revised as suggested. (Line 97)

L156-157: I would rephrase the definition of the continuous melt/freeze onset to the "the day after which ice surface melting/freezing conditions persist".
Reply: We modified the definition of continuous melt and freeze onset as suggested. (Line 160)

L224: There is no Table S1.
Reply: Table S1 is in the supplement and includes a summary of surface and basal melt/freeze onsets from different methods.

L241: Paragraph 3.1 (Comparison of ice surface melt and freeze onsets from different methods): The period of time covered by this analysis is unclear. You mention that you use 55 IMB trajectories but not the time period.
Reply: Thanks for your comment. We specified the time period ("2002 to 2018"). (Line 247)

L265: I would replace Figure 3 by a table since half of the entries are empty.
Reply: We prefer to keep this figure, since the colormap also shows the level of difference between the 4 pairs of surface melt and freeze onsets. We enlarged the figure for better visibility of the written text and specified the meaning of "+" and "–", according to other reviewers' suggestions. We hope for your understanding. (Figure 3)

L291: I would suggest using days instead of "d" here and throughout the manuscript
Reply: We changed "d" to "days" throughout the manuscript, as suggested.

L308-309: Please rephrase "FO is primarily controlled by the decline of net shortwave radiation as the approaching of polar night". The sentence is confusing, not clear.
Reply: We rewrote the sentence as "…, as the SFO is primarily controlled by the decline of net shortwave radiation with the approach of the polar night.". (Lines 320-321)

L 310: Table 2 is not referenced in the text.
L381: Table 4 is not referenced in the text.
L334: you mention Figure 5b but there is no previous mention of Figure 5a.
Reply: Good point. We now added references to Table 2, Table 4 and Figure 5a in the text. (Line 305; 389; 346-348)

L314: Can you be a little more specific than "~several mK"?
Reply: We now refer to the original reference, Shaw et al. (2009), and changed

"~several mK" to "within a few mK of the freezing point". (Lines 325)

L327: "absorbs" instead of "absorb".
Reply: Revised as suggested. (Line 339)

L331-332: I would rephrase "Here we further investigate the mechanism relevant to the BFO from the perspectives of both sea ice itself and underlying Ocean". It is unclear.
Reply: The sentence was rewritten as "We further investigate the mechanism relevant to the time lag between the BFO and SFO from the perspectives of both the sea ice itself and the underlying ocean.". (Line 345)

L356: Delete "here".
Reply: Deleted. (Line 370)

L435: I believe you are referring to Figure 4 instead of Figure 3?
Reply: Yes, good catch. Changed to "Figure 4c and 4d". (Line 449)

L501: Replace "ascribe" by "attributed".
Reply: Revised as suggested. (Line 519)

L502: replace "Lagrangion" by "Lagrangian".
Reply: Revised as suggested. (Line 518)

L507-507: I would rephrase" Second, the presences of ice interior melt" to "Second, interior ice melt".
Reply: Revised as suggested. (Line 525)

L516: Replace "varieties" by "diverse".
Reply: Revised as suggested. (Line 534)

**Response to RC3**

L32, add "," before "play a key role"
Reply: Added as suggested. (Line 32)

L37, L39, Please check if the references are cited properly.
Reply: Although there is no direct connection between the changes in the length of the melt/freeze season and sea ice deformation, the melt/freeze processes do alter the ice properties and hence the sea ice deformation. The Arctic-wide melt season has lengthened at a rate of 5 days decade$^{-1}$ from 1979 to 2013, dominated by later autumn freezeup within the Kara, Laptev, East Siberian, Chukchi, and Beaufort seas between 6 and 11 days decade$^{-1}$(Stroeve et al., 2014). So we consider theses as suitable

citations.

Persson, P. O. G.: Onset and end of the summer melt season over sea ice: Thermal structure and surface energy perspective from SHEBA, Clim. Dyn., 39(6), 1349-1371, doi:10.1007/s00382-011-1196-9, 2012.

L53, Add references for the statement in L50-53.
Reply: We added a reference here: Persson (2012). (Line 53)

L54, "step" or "step-like"?
Reply: We changed the "step" to "step-like". (Line 55)

L64, can remove "due to freezing" after "basal ice growth".
Reply: Removed as suggested. (Line 65)

L80, move "freezing and thawing processes" before "sea ice surface"-
Reply: Revised as suggested. (Lines 82-83)

L81, It seems that "Another method" appears very sudden. For this, L66-80 can be listed as a separate paragraph. In addition, please rewrite the sentences of L66-68, for example, "The sea ice freeze-thaw cycle can be identified using the data measured by sea ice mass balance buoys (IMBs), which consist of a thermitor chain in combination with … (Perovich et al. 2021)."
Reply: This part has been revised as suggested. (Line 68)

L110-114, rewrite the sentences to more concise, e.g., "The IMBs deployed on landfast ice are excluded because shallow coastal waters have … (Eicken et al., 2005)".
Reply: Changed as suggested. (Lines 114-116)

L116, "The surface melt and freeze onsets is related to observations of near surface air temperature collected by the IMB" or "The surface melt and freeze onsets is related to the near surface air temperature", Please rewrite this sentence.
Reply: We removed "collected by the IMB" here. (Line 118)

L144-146, If the 17 ITPs deployed in the central BG include the 12 ITPs used for detecting the basal melt or freeze?
Reply: 7 of the 12 ITPs used for detecting the basal melt or freeze were in the BG, and were included in the 17 ITPs. The other 5 ITPs in the CAO were not. The 17 ITPs deployed in the central BG are listed in Figure 9a, and the 12 ITPs used for detecting the basal melt or freeze are listed in Table 3.

L179, remove "of the" at the beginning of this line.
Reply: Removed as suggested. (Line 182)

Reply: The four pairs of surface melt and freeze onsets and one pair of basal melt and growth onsets have been listed in the supplement, due to their large space requirement.

Reply: Revised as suggested. (Lines 229-239)

Reply: Corrected as suggested. (Figure 2)

Reply: Removed as suggested. (Line 259)

Reply: Yes. the time when the SAT decreases below freezing corresponds to SFO-SAT. The sentence was rewritten as "… , with an average delay of 13 days after SFO-SAT." (Line 261)

Reply: We deleted "defined as" and changed "median" to "moderate". The "-" and "+" in tables is the time difference of column minus row, which we stated in the caption of the figure. (Line 267; Figure 3)

Reply: We checked the time lag between BFO and SFO again. The average SFO was on 20 August, and the average BFO was on 14 November. The BFO was determined 86 days after the SFO, which is almost three months. In addition, the median date of the BFO and SFO (has been added in the text according to another reviewer's suggestion) shows the same result. (Lines 277-280)

L276, confuse about "a decreasing trend of surface and basal melt onset" and "increasing trend of freeze onsets". Do you mean "surface and basal melt onsets later and freeze onsets earlier with an increase of latitude" or "surface and melt onsets are

decreasing and the freeze onsets are increasing with the increase of latitude? Please
rewrite.
Reply: We rewrote the sentence to read "The overall spatial patterns revealed a shift
in the surface and basal melt onsets to earlier dates while the freeze onsets shift to
later dates with a decrease in latitude, as one would expect.". (Lines 285-286)

L285, The meaning of "YD" in figure 4. For easy understanding, it is better to use
date and month, e.g., 11 June, also in the figure, not Julian Day. Similar for Figure 8,
and Figure 11.
Reply: We changed the time axis labels in Figure 4, Figure 8, and Figure 11 to
"MM/DD" as suggested, as well as throughout the manuscript.

L310, Please refer Table 2 in the text. The average surface net radiation changes were
calculated using ERA5 reanalysis data. Please add this information to the Table 2.
Reply: We added the reference to Table 2, and included the information in the title as
suggested. (Lines 305; 320-321)

L312-313, Difficult to understand this sentence, please rewrite it.
Reply: The sentence was rewritten as "We compare $\Delta T$, $u_{*0}$, $F_w$, and $F_c$ during the 10
days before (–10d) and after (+10d) our calculated BMO using data from 12 pairs of
co-located IMBs and ITPs (Table 3).". (Lines 323-324)

L314-315, "almost twice as large", compare to which?
Reply: We rewrote the sentence as "… almost twice as large as $\Delta T$ (–10d),…" (Line
326)

L324-329, The trigger of SMO is an increase of longwave radiation (L300-301), and
the key to BMO is when Fw becomes greater than Fc (L322). Thus the explanation in
this paragraph only can explain why the BMO in the BG occurred much earlier than
the BMO in the CAO, but cannot explain why the BMO in the BG much earlier than
the SMO, while they occurred almost at the same time in the CAO.
Reply: The SMO and BMO depend on the heat budget of the ice surface and base,
respectively. In addition to the larger amount of incoming solar radiation based on the
latitude, the larger fraction of leads also results in more absorption of solar radiation.
The last sentence in this paragraph used "may" and "partly" as a speculation.

L346, change "," to ";" between two references.
Reply: Changed as suggested. (Line 360)

L347-348, can be more clear or concise as "jointly resulted in the time of BFO later
approximately 3 months than SFO.".
Reply: Changed as suggested. (Line 362)

L361, "plays the opposite role of a thermal insulator" seems wrong. Please rewrite it.

Reply: We agree that this statement does not read as intended. This has been rewritten to "The lower SAT accelerates the sea ice cooling, while a thicker snow cover as a thermal insulator plays the opposite role due to its low thermal conductivity (Ledley, 1991).". (Lines 374-376)

L375, "Figure 5" should be "Figure 6".
Reply: This has been changed. (Line 389)

L382, refer Table 4 in the text. The meaning of ΔHi(m) in the Table?
Reply: Table 4 is now referred in the text and the column of ΔHi(m) was deleted. (Line 389; Table 4)

L388, use different symbols for the surface and ice melt, to discriminate the symbols used in Eq. (5).
Reply: We changed $H_i$ and $H_s$ to $\Delta H_i$ and $\Delta H_s$ according to another reviewer's comment. (Lines 403-404)

L396, remove "had"
Reply: Removed as suggested. (Line 411)

L408, the meaning of (2.14m), (0.77m), (0.22m) ?
Reply: These numbers refer to the total basal melt.

L410, what does the "initital" mean in "the initial ice thickness"? It means the ice thickness when the IMB installed, or the ice thickness when the SMO commences?
Reply: "initial ice thickness" refers to the time when basal melt begins, i.e., the BMO. We specified this in the text as "… initial ice thickness when basal melt begins". (Line 425)

L412-413, Figure 6 indicate the linear relationship between ICI and BFO-SFO, which didn't consider the ice concentration. Thus this statement cannot relate to Figure 6 as stated "relative to the linear regression as shown in Figure 6".
Reply: This statement has been removed.

L421, add "Thus" before "The average annual ice thickness" to connect this sentence with the previous one.
Reply: The change was made as suggested. (Line 436)

L432, the caption of (d) and the figure ylabel is not same. Please correct the wrong one.
Reply: The change was made as suggested. (Line 446)

L435, Should "Figure 3c and 3d" be "Figure 4c and 4d"? recheck.

Reply: We changed this to "Figure 4c and 4d". (Line 449)

L443-444, The positive feedback between thinner and more vulnerable ice and earlier BMO is a general statement. How can you derive this feedback from L435- 443 where you talked about the decal changes of BMO and the average oceanic mixed layer temperature departure from the local freezing point.
Reply: We agree and deleted this sentence.

L453: I don't agree "the IMB observations do not catch the freezing and thawing of the seasonal sea ice". This is also paradoxical your method for detecting SMO, SFO, BMO and BFO from IMB observations.
Reply: The key point of this sentence is to emphasize "seasonal/first-year sea ice", since the majority of IMB were deployed on second year/multiyear ice. That is the reason why we also used the ULS data to represent the seasonal sea ice. We rewrote the sentence to emphasize the different focus between IMB and ULS observations. (Lines 466-468)

L468, "by the two methods" or "by the two moorings"? From the text, it seems that you mean "by the two moorings".
Reply: The two methods refer to the two data sources, i.e., IMB and ULS. We changed this now to read "…obtained from IMBs and ULS revealed…". (Lines 482-483)

L480: Can remove BGOS-C: cyan since have excluded this mooring.
Reply: We have removed BGOS-C from Figure 10 and Figure 11.

L498-499, Difficult to understand this sentence, please rewrite it.
Reply: We rewrote the sentence as "…, with lower surface air temperature, thinner sea ice, and thinner snow cover favoring earlier onset of basal ice growth, and vice versa.". (Line 524)

---

## Author Response (AR2)

Dear Dr. Derksen

We would like to submit our revised manuscript entitled "Changes in the annual sea ice freeze-thaw cycle in the Arctic Ocean from 2001 to 2018" [tc-2022-137], which we submitted to the Cryosphere. Thank you for your comments. We completed all the editorial corrections as your suggestions.

Thank you for your time.
Sincerely,
Ruibo Lei, and other co-authors